# SPADE: S̲EMANTIC-P̲RESERVING A̲DAPTIVE D̲ETOXIFICATION OF IMAGES

## ABSTRACT

Image generation models often struggle with safety-critical edits, especially detoxifying harmful visual content without losing semantic context. We introduce SPADE, a novel dataset for *controlled, graded detoxification of toxic images*. Each toxic image is paired with three semantically aligned, progressively detoxified variants that preserve knowledge relevance, scene context, and visual consistency. This enables models to learn nuanced, fine-grained detoxification editing beyond binary filtering, addressing the trade-off between harm reduction and semantic preservation. SPADE comprises 2,500 toxic images and multi-level detoxified counterparts, captions, and contextual stories. We benchmark detoxification through human preferences, CLIP-based similarity, and structural metrics, establishing SPADE as the first resource for graded, controllable detoxification in image generation. Our work lays the foundation for safe, interpretable, and context-aware visual moderation. Our code and data are publicly available[1].

## 1 INTRODUCTION

The rise of online platforms (Facebook, Instagram, X) has accelerated the spread of toxic, violent, or hateful imagery, often reused without moderation (Hu et al., 2025). Meanwhile, large-scale text-to-image (T2I) models (e.g., Stable Diffusion (Rombach et al., 2022a)) trained on web-scale data inherit cultural, ideological, and representational biases and can easily produce harmful content (Kim et al., 2025; Wu et al., 2024) even using simple prompts (Qu et al., 2023; Wu et al., 2025). Such misuse (including cyberbullying and hate imagery) poses urgent challenges for safe AI (Hee et al., 2024; Khullar et al., 2025).

Prior efforts on prompt engineering for safe T2I generation (Feng et al., 2024; Schramowski et al., 2023a; Gandikota et al., 2023) and instruction-based editing (such as InstructPix2Pix and localized semantic editing methods) (Brooks et al., 2023a; Yang et al., 2024; Bai et al., 2024) mitigate toxicity only partially. They often fail at preserving semantics, context, and realism (Khanuja et al., 2024), largely due to the lack of structured training datasets providing multi-level detoxified image variants with coherent contextual narratives.

We address this gap with SPADE, a new dataset for semantic-preserving, graded detoxification. Each toxic image is paired with three progressively detoxified variants, guided by captions and contextual stories. As shown in Fig. 1, this design enables nuanced editing: e.g., transforming "A hand holding a bloody knife" into variants with a knife with fruits, a toy knife, and finally a paint knife, each safer but semantically coherent. The second example shows a toxic image of "A man and a woman arguing on the street" progressively detoxified into "having a heated discussion," "talking loudly," and finally "conversing," while preserving semantics and context. Both examples showcase the core challenge of **semantic-preserving image-to-image (I2I) detoxification**. Unlike binary filtering of toxic images (Gomez et al., 2020; Das et al., 2023), our approach mirrors real-world moderation, allowing dynamic adjustment of toxicity levels (e.g., for "Safety Grade 2" users).

Our contributions are: (1) **Task**: We propose a novel multimodal detoxification task: reducing toxicity in images while preserving semantics via image+story conditioned editing. (2) **Dataset**: We release SPADE, a novel dataset containing 2,500 real-world toxic images with captions, contextual stories, and three semantic-preserving detoxified synthetic variants each. (3) **Metrics**: We de-

---

[1] https://anonymous.4open.science/r/ICLR_detoxic_image-1BD4/

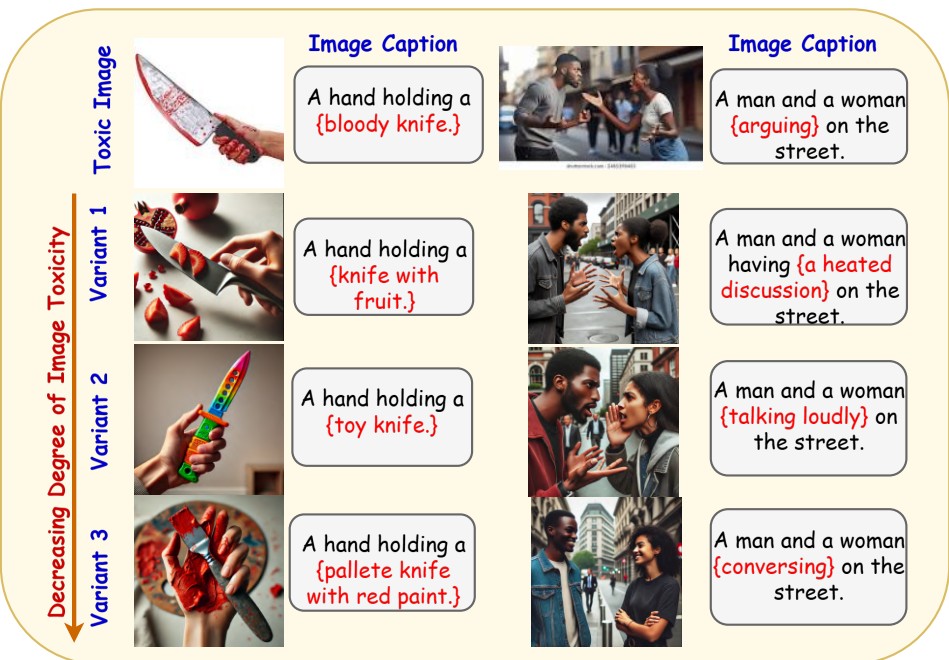

Figure 1: Two examples from our SPADE dataset for controlled detoxification of toxic images. Each toxic image (top row) is paired with three progressively detoxified variants $\mathcal{V}_1$-$\mathcal{V}_3$, generated via story-guided image-to-image conditioning. These stories embed semantically aligned, detoxified captions that guide the transformation while preserving contextual and narrative coherence. The captions illustrate the lexical softening across toxicity levels, for graded multimodal supervision.

sign comprehensive evaluation metrics to measure the fidelity across dimensions, such as semantic preservation, knowledge relevance, visual realism, and story alignment. (4) **Baseline**: We fine-tune ControlNet (Zhang et al., 2023b) for sequential detoxification, showing strong reductions in toxicity with preserved semantics, visual fidelity, and narrative consistency, providing the first baseline for this task.

## 2 RELATED WORK

**Image Generation:** Image generation has advanced across paradigms such as image-to-image translation (Mao et al., 2018; Andreini et al., 2021), sketch-to-image (Lu et al., 2018), conditional and few-shot synthesis (Esser et al., 2018; Cloûatre & Demers, 2019), layout/pose-guided methods (Zhu et al., 2019; Ma et al., 2018), and panoramic creation (Tripathi et al., 2019). T2I generation is now dominant: early GAN-based models (Xu et al., 2018; Sharma et al., 2018) gave way to diffusion models with state-of-the-art fidelity and controllability (Rombach et al., 2022a). Large-scale systems such as DALL-E (Ramesh et al., 2022), Imagen (Saharia et al., 2022), and Stable Diffusion (Podell et al., 2023) leverage billions of web pairs, but also propagate dataset biases (Kim et al., 2025). Recently, Sora (Liu et al., 2024) extended diffusion to high-quality text-to-video generation.

**Controllable and Safe Generation**: Despite progress, T2I systems often produce toxic or biased content (Weidinger et al., 2022; Schramowski et al., 2022). Existing safety mechanisms like prompt filtering, red-teaming, post-hoc moderation, offer limited control and risk over-/under-correction. Editing approaches like InstructPix2Pix (Brooks et al., 2023b) and Prompt-to-Prompt (Hertz et al., 2022) enable local, semantic modifications, while diffusion models introduce control via layouts, conditioning, or guidance networks (Rombach et al., 2022b; Avrahami et al., 2023). Plug-and-play refinements (Tumanyan et al., 2022; Mokady et al., 2022) and disentangled frameworks (Gafni et al., 2022) highlight new avenues for controllable, ethically tuned generation, though toxicity reduction remains underexplored.

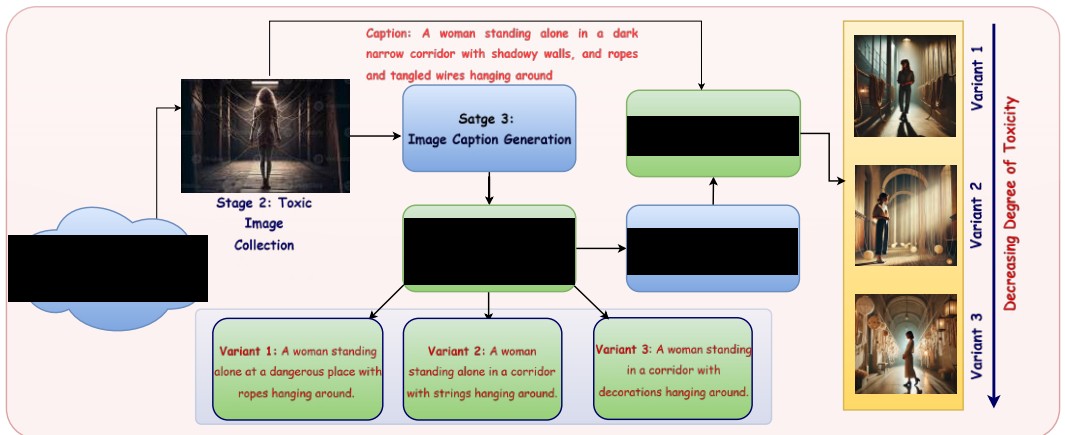

Figure 2: An overview of our **SPADE** dataset creation pipeline. The process is organized into six sequential stages as detailed in Section 3.

**Image Detoxification and Semantic Editing**: Recent methods emphasize personalization or stylistic editing, including inpainting (Zhang et al., 2020; Avrahami et al., 2022), semantic fusion (Andonian et al., 2023), and fine-tuning approaches such as Imagic and DreamBooth (Kawar et al., 2023; Ruiz et al., 2023). However, these prioritize fidelity and expressiveness over ethical detoxification, remain costly, and are vulnerable to circumvention (Schramowski et al., 2023b). Crucially, no existing framework supports progressive, semantic-preserving detoxification. Our work introduces the first dataset and task enabling step-wise toxicity reduction through multimodal conditioning, addressing this gap for safe, controllable visual moderation. Detailed related work is in Appendix A.

## 3    SPADE DATASET CURATION PIPELINE

We design a multi-phase pipeline (Fig. 2) to construct a high-quality dataset for toxicity-aware visual grounding and detoxification.

**Stage 1: Toxic Keyword Collection.** Inspired by LAION (Schuhmann et al., 2021), we compile ~200 toxic keywords via GPT-4[2] using Azure AI's harm taxonomy[3] (violence, hate, sexual, self-harm) as seed inputs, and the prompt: "Give me 50 queries to search for images with similar kinds of harm/offense." Table 5 in Appendix B.1 shows the full harmful keywords list. This lexicon guides large-scale web image retrieval to ensure that the resulting dataset reflects the distribution and public availability of real-world images.

**Stage 2: Toxic Image Collection.** Using Google Image Search[4], we retrieve the top 10 results per keyword and manually filter them for relevance, yielding 2,500 harmful images spanning diverse visual themes, content types, and harm modalities. See Appendix D for annotator details.

**Stage 3: Image Caption Generation.** To preserve the initial textual grounding for each toxic image, we employ LLaVA (Liu et al., 2023) to generate semantically rich captions $\mathcal{T}^k$ for each image $\mathcal{I}^k$, using the prompt: "Given this image $\mathcal{I}^k$, please provide an accurate caption $\mathcal{T}^k$ that explains the scene." This is followed by manual verification for semantic alignment, fluency, and relevance. These generated captions facilitate subsequent conditioning and generation tasks. Appendix E shows examples with quality scores.

**Stage 4: Detoxified Caption Variant Generation.** GPT-4 rewrites each toxic caption $\mathcal{T}^k$ into three progressively detoxified variants $\{\mathcal{V}_i^k\}_{i=1}^3$, preserving semantics while reducing harmfulness. Prompt is in Appendix C.

**Stage 5: Contextual Story Construction.** Existing image editing models often fail to incorporate coherent narrative context, resulting in outputs that lack semantic grounding or continuity with the original image content (Park et al., 2025; Xia et al., 2025). Hence, to embed captions in coherent

---

[2] https://openai.com/index/gpt-4-research/   [3] https://learn.microsoft.com/en-us/azure/ai-services/content-safety/
[4] https://images.google.com/

narratives, GPT-4 generates short 10-sentence stories $\mathcal{S}^k$ around each $(\mathcal{I}^k, \mathcal{T}^k)$, ensuring temporal and contextual grounding. Prompt is in Appendix C. This ensures that the generated story explicitly incorporates the caption $\mathcal{T}^k$ while situating the image within a coherent and contextually meaningful narrative. We manually evaluated 100 such stories on a 1–5 quality scale, assessing spatial relevance, emotional plausibility, and narrative coherence, and found them to be high quality (see Appendix F). Appendix G shows example stories including the one for the example in Fig. 2.

**Stage 6: Final Detoxified Image Variant Generation.** Finally, DALL-E 3 (Shi et al., 2020) generates detoxified image variants conditioned on toxic images, detoxified captions (integrated into the stories), and stories, enabling paired multimodal supervision. We conduct a qualitative evaluation of the generated detoxified images, focusing on context preservation and toxicity reduction. See Appendix I for detailed samples and evaluation protocol.

## 4 SPADE DATASET ANALYSIS

The proposed SPADE dataset contains 2500 images each with 3 detoxified variants, captions and a story. Toxicity score for toxic images is 79.7 while that for variants $\mathcal{V}_1$-$\mathcal{V}_3$ are 73.4, 54.6 and 35.8 respectively[5]. This shows that progressive toxicity reduction is effectively captured in the dataset.

### 4.1 QUALITY EVALUATION

**Evaluation Metrics.** We evaluate quality of the detoxified image variants compared to original toxic images in SPADE on four core dimensions: knowledge relevance (KR), context preservation (CP), semantic similarity (SS), and structural fidelity (using Fréchet Inception Distance or FID).

KR evaluates factual consistency of detoxified image variants relative to their toxic reference images using o4-mini[6] as an automated judge. The model scores each detoxified image on a 5-point Likert scale (1–5) (Likert, 1932) based on how well it preserves the real-world semantics and core informational content (core entities, actions, settings, and concepts) of the reference, with higher scores indicating stronger factual alignment. This metric focuses strictly on knowledge-level correctness rather than aesthetics. Prompt is in Appendix C.

CP evaluates whether a detoxified image maintains the original meaning, communicative intent, emotional tone, and narrative structure of its toxic reference. Using o4-mini as an automatic evaluator, each detoxified–reference image pair is scored on a 5-point Likert scale (1 = completely lost, 5 = completely preserved). The metric emphasizes continuity of message, tone, humor, and implied narrative, ignoring superficial stylistic differences unless they affect interpretation.

SS measures semantic consistency using CLIP (Radford et al., 2021), computing cosine similarity between embeddings of toxic and detoxified images. Higher scores indicate stronger preservation of core meaning despite toxicity reduction.

Perceptual fidelity (FID) is assessed with SSIM (Wang et al., 2004), PSNR (Huynh-Thu & Ghanbari, 2008), and LPIPS (Zhang et al., 2018). SSIM captures structural coherence (luminance, contrast, and texture), PSNR measures signal fidelity, and LPIPS evaluates perceptual similarity in deep feature space. Together, they provide a robust measure of realism and semantic preservation.

**KR and CP for SPADE.** Variant $\mathcal{V}_1$, which shows minimal detoxification, achieves the highest scores in both KR (3.75) and CP (2.81), closely aligning with the original image in terms of context content and scene structure. Variants $\mathcal{V}_2$ and $\mathcal{V}_3$ show a progressive decline in both metrics (KR: 3.37 and 3.06, CP: 2.54 and 2.15), illustrating the expected trade-off between reducing toxicity and maintaining original semantics. Despite this decline, the scores indicate that even the most detoxified variant ($\mathcal{V}_3$) retains a substantial degree of relevance and contextual grounding. These results confirm the effectiveness of our proposed framework in preserving essential semantic attributes while enabling controlled detoxification, thus validating the utility of SPADE for developing robust and interpretable multimodal content moderation systems.

Table 1 shows CP and KR scores for a sample. The reference image shows a hostile protest, and the subsequent variants show progressive detoxification by altering textual content on placards while

---

[5] Calculated using a CLIP classifier finetuned on dataset from (Hendrycks et al., 2021)
[6] https://openai.com/index/introducing-o3-and-o4-mini/

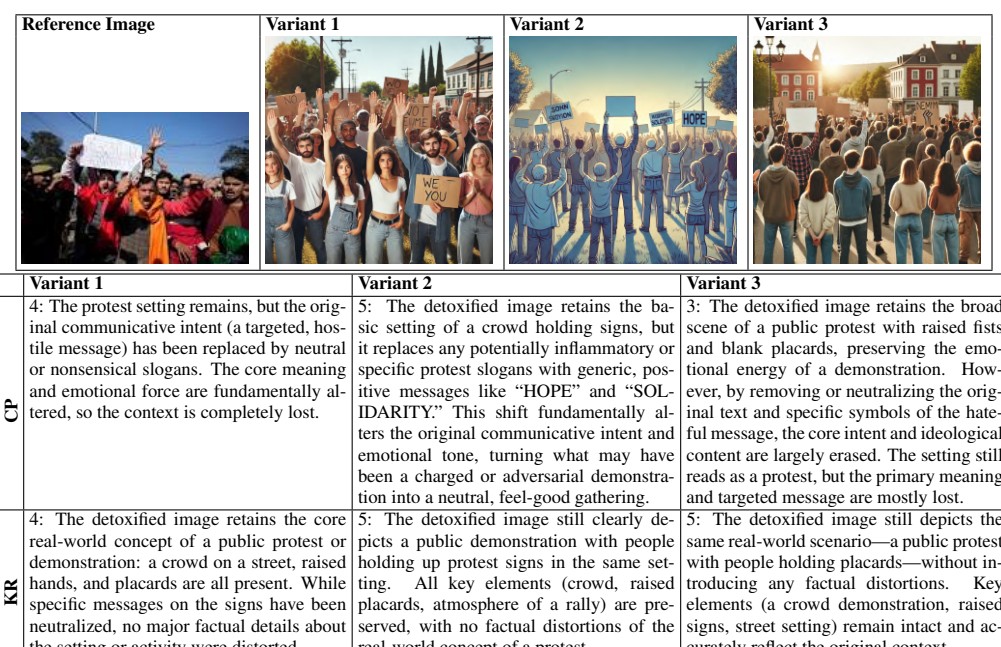

| | Variant 1 | Variant 2 | Variant 3 |
|---|---|---|---|
| **CP** | 4: The protest setting remains, but the original communicative intent (a targeted, hostile message) has been replaced by neutral or nonsensical slogans. The core meaning and emotional force are fundamentally altered, so the context is completely lost. | 5: The detoxified image retains the basic setting of a crowd holding signs, but it replaces any potentially inflammatory or specific protest slogans with generic, positive messages like "HOPE" and "SOLIDARITY." This shift fundamentally alters the original communicative intent and emotional tone, turning what may have been a charged or adversarial demonstration into a neutral, feel-good gathering. | 3: The detoxified image retains the broad scene of a public protest with raised fists and blank placards, preserving the emotional energy of a demonstration. However, by removing or neutralizing the original text and specific symbols of the hateful message, the core intent and ideological content are largely erased. The setting still reads as a protest, but the primary meaning and targeted message are mostly lost. |
| **KR** | 4: The detoxified image retains the core real-world concept of a public protest or demonstration: a crowd on a street, raised hands, and placards are all present. While specific messages on the signs have been neutralized, no major factual details about the setting or activity were distorted. | 5: The detoxified image still clearly depicts a public demonstration with people holding up protest signs in the same setting. All key elements (crowd, raised placards, atmosphere of a rally) are preserved, with no factual distortions of the real-world concept of a protest. | 5: The detoxified image still depicts the same real-world scenario—a public protest with people holding placards—without introducing any factual distortions. Key elements (a crowd demonstration, raised signs, street setting) remain intact and accurately reflect the original context. |

Table 1: CP and KR scores with justification for each variant for an example from SPADE.

| | R | $V_1$ | $V_2$ | $V_3$ |
|---|---|---|---|---|
| R | 1.00 | 0.95 | 0.70 | 0.60 |
| $V_1$ | 0.95 | 1.00 | 0.65 | 0.50 |
| $V_2$ | 0.70 | 0.65 | 1.00 | 0.55 |
| $V_3$ | 0.60 | 0.50 | 0.55 | 1.00 |

Table 2: CLIP-based pairwise cosine similarity between captions of toxic image and detoxified variants ($V_1$–$V_3$).

preserving the scene structure (e.g., crowd, raised hands). The KR scores remain high across all variants ($\geq 4$), demonstrating that factual details such as the setting, activity type, and participant behavior are retained. CP, however, reveals controlled moderation: from aggressive slogans to neutral, then to symbol-less placards.

**Assessing Semantic Similarity in SPADE.** Table 2 reports CLIP (ViT-B/32) cosine similarity between captions of toxic image and detoxified variants ($V_1$–$V_3$). Results show a trade-off: $V_1$ stays close to the reference, while $V_2$ and $V_3$ diverge to ensure safety. Still, moderately high scores indicate that core context and narrative intent are preserved despite progressive detoxification.

## 4.2 t-SNE Analysis of SPADE.

To qualitatively analyze semantic behavior across detoxification stages, we examine the t-SNE (van der Maaten & Hinton, 2008) projection of CLIP embeddings for a representative base toxic sample of 100 images, and their three detoxified variants. As shown in Figure 5, the original toxic samples (●) serve as the semantic anchor, typically positioned at the periphery of the embedding space due to their heightened toxicity and distinct features. As detoxification begins, variant $\mathcal{V}_1$ (■) initiates a directional shift away from the base, reflecting early-stage mitigation of toxic traits while retaining core semantics. Variant $\mathcal{V}_2$ (◆) continues this trajectory, moving further from the base and closer to the detoxified cluster. It demonstrates more pronounced semantic refinement, often involving structural rephrasing or contextual rebalancing. Finally, variant $\mathcal{V}_3$ (▲) exhibits the most compact clustering, indicating convergence toward a detoxified yet semantically coherent representation. The smooth progression highlights effective toxicity reduction while preserving semantic integrity.

| Datasets | Real Image | Automatic Generated | Editing Region | Open Domain | Source Example | Instruction | Target Example |
|---|---|---|---|---|---|---|---|
| EditBench | ✓ | ✗ | ✓ | ✓ |  | Change emotion: $Disgust \rightarrow Sad$ |  |
| MagicBrush | ✓ | ✗ | ✓ | ✗ |  | make the man ride a motorcycle |  |
| HQ-Edit | ✗ | ✓ | ✗ | ✓ |  | Alter her hair color to black. |  |
| Instruct Pix2Pix | ✗ | ✓ | ✗ | ✓ |  | Add fireworks to the sky |  |
| UltraEdit | ✓ | ✓ | ✓ | ✓ |  | Add a moon in the sky |  |
| HumanEdit | ✓ | ✗ | ✓ | ✓ |  | Convert a suitcase into a dog. *Action: Replace* |  |
| *Transcreate Images* | ✓ | ✓ | ✗ | ✗ |  | a dish of food with beef and vegetables on it. *Target: United States* |  |
| *Ours (SPADE)* | ✓ | ✓ | ✗ | ✓ |  | Story |  Variant 1  Variant 2  Variant 3 |

Table 3: Comparison of our SPADE dataset with existing text-to-image editing benchmarks. Unlike prior datasets, SPADE combines real-world toxic images with automatically generated detoxified variants across controlled toxicity levels. It supports semantically aligned editing with grounded instructions and target examples, tailored specifically for safety-critical issues.

## 4.3 COMPARISON WITH OTHER DATASETS

Table 3 presents a detailed comparison of our SPADE dataset with the existing T2I editing benchmark datasets. Unlike other datasets such as EditBench (Lin et al., 2024) and MagicBrush (Zhang et al., 2023a), which rely solely on real-world images and manual editing, or HQ-Edit (Hui et al., 2024) and InstructPix2Pix (Brooks et al., 2023b), which generate synthetic data via diffusion-based augmentation, SPADE is the only dataset that (i) integrates real-world toxic content with (ii) automatically generated detoxified image variants grounded in multimodal contextual storytelling. Furthermore, existing datasets typically focus on stylistic, aesthetic, or attribute-based modifications (e.g., color changes, object replacement), and often overlook the complexity of ethical, multimodal visual cues. In contrast, SPADE embeds each image within a rich story context and controls generation using LLM-derived instructions that semantically reinterpret the harmful content while retaining scene coherence. Furthermore, SPADE introduces a structured multi-step detoxification pipeline with three graded visual variants ($V_1$ to $V_3$), a feature not found in any existing dataset. While image transcreation (Khanuja et al., 2024) focuses on cultural translation of food items or scenes, our SPADE dataset targets ethically grounded adaptations of offensive/harmful images across safety-critical domains.

## 5 Image Detoxification Method

To enable controlled detoxified image generation, we fine-tune a pre-trained T2I model, **ControlNet** (Mou et al., 2024), using instruction-guided image-to-image generation. This approach allows us to apply targeted modifications, such as removing sensitive elements or recontextualizing scenes, while preserving both visual fidelity and semantic coherence. Instead of fine-tuning over the entire dataset in one pass, we introduce a **sequential fine-tuning strategy** (Hu et al., 2024; Guan et al., 2025) over our curated SPADE corpus, allowing the model to learn progressively detoxified transformations in a graded manner. Let $\mathcal{I}$ be the original (or base) toxic image with its associated caption $\mathcal{T}_0$ and three incrementally detoxified image-caption pairs $(\mathcal{V}_k, \mathcal{T}_k)$, where $\mathcal{T}_k$ articulates the revised narrative associated with $\mathcal{V}_k$. Caption $\mathcal{T}_k$ is embedded into story $\mathcal{S}$ to obtain revised story $\mathcal{S}_k$.

**(i) Stage 1 (Variant $\mathcal{V}_1$ Generation):** We fine-tune ControlNet on image-text pairs $(\mathcal{V}_1, \mathcal{S}_1)$, using the original toxic image $\mathcal{I}$ as the **reference image** for conditioning. This helps maintain the context and background similarity between $I$ and $\mathcal{V}_1$.

**(ii) Stage $k$ (Variant $\mathcal{V}_k$ Generation):** Subsequently, for each additional variant $(\mathcal{V}^k, \mathcal{S}^k)$, we continue fine-tuning the model using $\mathcal{V}^{k-1}$ (the *previous* less-toxic variant) as the **reference image**. This iterative conditioning ensures a gradual decrease in toxicity while meticulously preserving the context and knowledge relevance from the preceding, less-toxic version. By sequentially fine-tuning ControlNet on such variant datasets, we introduce a **conditioning approach**. This method conditions the pre-trained T2I models on reference images without requiring any architectural modifications to the base T2I model or ControlNet itself. This technique empowers textual instructions to effectively control the target structure and content, while simultaneously ensuring the preservation of the original image's context and overall visual integrity.

We train on 2000 samples from SPADE (with a small subset used for validation) and test on the remaining 500 samples.

## 6 Results and Analysis

To evaluate the effectiveness of our proposed Sequential ControlNet model, we present a comparative performance analysis against several strong baselines: (i) Stable Diffusion (Zero-shot), (ii) ControlNet (Zero-shot), (iii) Stable Diffusion (Fine-tuned), (iv) ControlNet (Fine-tuned), (v) Safe Diffusion (Schramowski et al., 2023b) and (vi) Sequential ControlNet (Proposed). A comprehensive description of each baseline architecture is provided in Appendix J.

**Automatic Results.** To evaluate the effectiveness of detoxified image generation, we conduct a detailed quantitative analysis leveraging two widely accepted metrics: CLIP-based cosine similarity (CosSim) for semantic alignment and FID (averaged over all 3 variants as compared to original toxic image[7].) for visual fidelity. For Stable Diffusion, we perform only Variant 1 generation since it can take only text (and no image) as conditioning input. Results in Table 4 show that our proposed model (Sequential ControlNet) achieves the highest CosSim across all variants, indicating strong semantic alignment between detoxified and original images. Concurrently, it yields the lowest FID, suggesting superior visual fidelity and distributional realism. In contrast, models such as Stable Diffusion (Zero-shot) and ControlNet (Zero-shot) show lower CosSim and higher FID. ControlNet (Finetuned) performs moderately well, aligning with its intermediate FID and cosine scores. Overall, the automatic metrics substantiate our model's ability to balance semantic consistency and perceptual quality, outperforming baselines (refer to the Table in the Appendix for variant-wise FID score).

**CLIP Embedding Visualization of Detoxification Levels.** To further assess semantic coherence and detoxification trajectories across models, we conducted a t-SNE analysis of image CLIP embeddings. As shown in Fig. 3, our proposed model (Sequential ControlNet) exhibits the most compact and semantically aligned clusters, with smooth transitions across variants, indicating consistent detoxification while preserving core semantics. In contrast, ControlNet (Zero-shot) and Stable Diffusion (Zero-shot) show broader dispersion and weaker alignment, suggesting that detoxification occurs with diminished semantic fidelity. Notably, ControlNet (Fine-tuned) demonstrates moderate clustering but reveals disjointed variant transitions, implying uneven detoxification effectiveness.

---

[7] Variant wise FID scores are in Appendix L

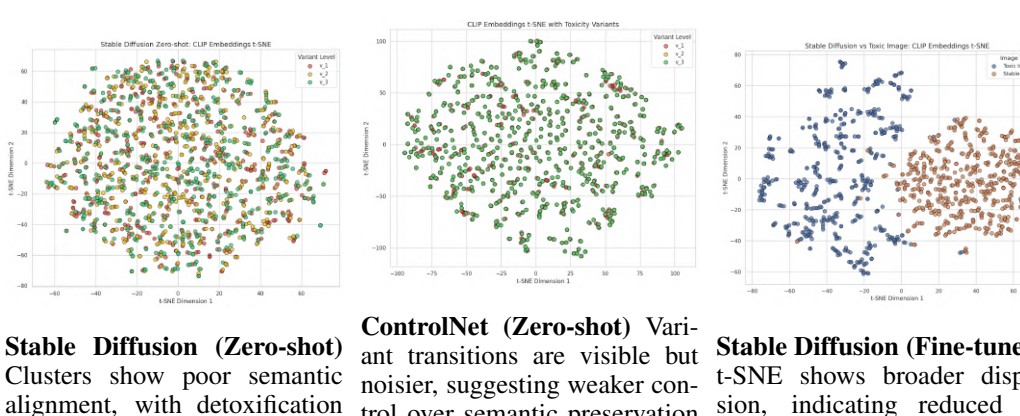

**Stable Diffusion (Zero-shot)** Clusters show poor semantic alignment, with detoxification trajectories over all the variants (V1, V2, and V3)

**ControlNet (Zero-shot)** Variant transitions are visible but noisier, suggesting weaker control over semantic preservation during detoxification.

.

**Stable Diffusion (Fine-tuned)**. t-SNE shows broader dispersion, indicating reduced semantic alignment compared to multi-variant models.

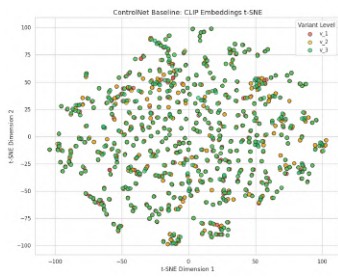

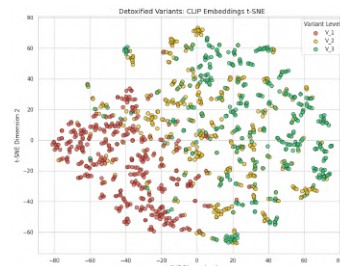

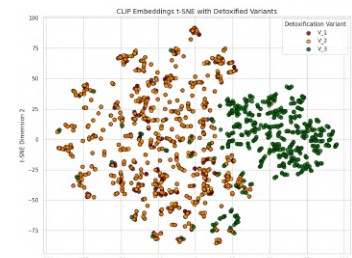

**ControlNet (Finetuned)** Semantic clusters are preserved across toxicity levels, with directional shifts from $v_3$ to $v_1$ indicating detoxification, but it is effective across variants.

**Safe Diffusion**. Despite missing samples, clusters remain coherent, with variant-level transitions visible. Improved semantics across variants

**Sequential ControlNet (Proposed)** shows tighter clustering and smoother trajectories, improved semantic consistency across detoxified variants.

Figure 3: t-SNE visualizations of CLIP image embeddings across six detoxification models. Each plot illustrates semantic structure and variant-level transitions for our test set.

| Model | Cosine Similarity ↑ | | | | | | FID ↓ | CP | | | KR | | |
|---|---|---|---|---|---|---|---|---|---|---|---|---|---|
| | $\mathcal{V}_1$-$\hat{\mathcal{V}}_1$ | $\mathcal{V}_2$-$\hat{\mathcal{V}}_2$ | $\mathcal{V}_3$-$\hat{\mathcal{V}}_3$ | $\mathcal{I}$-$\hat{\mathcal{V}}_1$ | $\mathcal{I}$-$\hat{\mathcal{V}}_2$ | $\mathcal{I}$-$\hat{\mathcal{V}}_3$ | | $\hat{\mathcal{V}}_1$ | $\hat{\mathcal{V}}_2$ | $\hat{\mathcal{V}}_3$ | $\hat{\mathcal{V}}_1$ | $\hat{\mathcal{V}}_2$ | $\hat{\mathcal{V}}_3$ |
| Stable Diffusion (Zero-Shot) | 0.565 | 0.572 | 0.572 | 0.518 | 0.520 | 0.523 | 240 | 3.86 | 4.09 | 3.87 | 4.79 | 4.82 | 4.51 |
| Stable Diffusion (Finetuned) | 0.583 | - | - | 0.514 | - | - | 271 | 2.05 | - | - | 2.35 | - | - |
| Safe Diffusion | 0.662 | 0.618 | 0.574 | 0.636 | 0.642 | 0.528 | 233 | 4.35 | 4.12 | 4.03 | 4.53 | 4.51 | 4.47 |
| ControlNet (Zero-Shot) | 0.662 | 0.665 | 0.649 | 0.639 | 0.640 | 0.640 | 205 | 4.22 | 4.14 | 4.13 | 4.98 | 4.98 | 4.97 |
| ControlNet (Finetuned) | 0.708 | 0.707 | 0.693 | 0.645 | 0.643 | 0.644 | 187 | 4.40 | 4.39 | 4.29 | 4.98 | 4.98 | 4.96 |
| Sequential ControlNet (Proposed) | 0.712 | 0.713 | 0.776 | 0.635 | 0.629 | 0.530 | 154 | 4.40 | 4.51 | 4/42 | 5.00 | 5.00 | 4.99 |

Table 4: Comparison of detoxified image quality across models.

Safe Diffusion maintains coherent clusters despite missing samples, while Stable Diffusion (Finetuned) shows the least semantic preservation, with embeddings scattered across the latent space. These findings reinforce the efficacy of our proposed model in achieving controlled detoxification with minimal semantic drift.

**Content Preservation (CP) and Knowledge Retention (KR).** As shown in Table 4, our proposed method outperforms all other approaches with consistently higher scores in both CP and KR across all variants. These results indicate near-perfect retention of both visual and contextual semantics post-detoxification. In contrast, ControlNet Finetuned and ControlNet Zero-Shot also yield strong performance, but with slightly lower KR values and less consistency in CP, especially in subtle visual shifts (e.g., $\mathcal{V}_3$). Safe Diffusion maintains competitive KR scores ($\approx4.5$) but exhibits variability in CP. Notably, Stable Diffusion Finetuned underperforms significantly in both metrics, especially with

KR dropping to 2.35 and CP to 2.05 in V1, confirming its limitations in generating semantically faithful detoxified outputs. These findings corroborate the t-SNE visual distribution and cosine similarity trends, which also revealed tighter clusters and higher semantic alignment for the proposed model. Collectively, the high CP and KR scores highlight the superiority of our sequential approach in producing detoxified images that preserve the original context without semantic drift.

**Case Studies.** In Fig. 4, we compare detoxification outputs from three models: Model 1 (Stable Diffusion with story input), Model 2 (ControlNet with image and story), and our Model 3 (Sequential ControlNet with story-image conditioning). For Example 1, Model 1 produces incoherent, cartoonish images, while Model 2 sometimes amplifies toxicity (e.g., adding fire). In contrast, Model 3 achieves smooth detoxification: variant 1 softens the street scene, variant 2 introduces a peaceful crowd, and variant 3 depicts a ceremonial parade, preserving semantics and reducing harm. In Example 2, Model 1 fails to stay relevant, and Model 2 generates even more toxic imagery. Model 3, however, transitions from symbolic dissociation to a benign soft toy, fully mitigating toxicity. These results highlight Model 3's superior ability to deliver smooth, context-preserving detoxification.

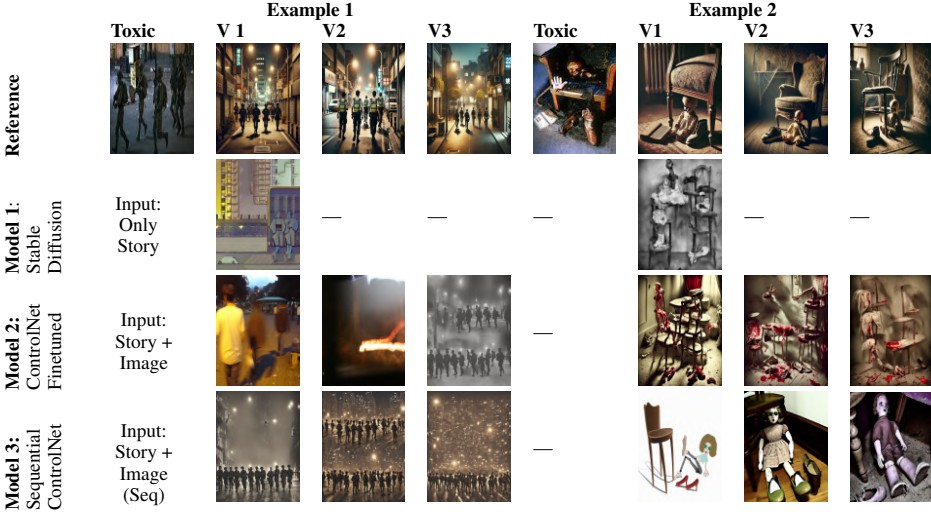

Figure 4: Variant-wise detoxification comparison across models for two representative examples. Our proposed model (Model 3) demonstrates smoother semantic transitions and stronger contextual preservation across detoxification stages.

**Error Analysis** While our sequential ControlNet model performs well overall, it has notable limitations. Key errors include: (1) Hallucination: Generates irrelevant or exaggerated elements not present in the input. See Fig. 6 and Appendix M.1. (2) Contextual Drift: Alters essential scene details (like subject positioning, object presence, or environmental cues), losing core semantics. See Table 11 and Appendix M.2. (3) Instruction Misinterpretation: Misaligns with nuanced or complex story guidance, particularly when the story involves nuanced semantic shifts or compound directives. See Table 12 and Appendix M.3. (4) Visual Artifacts: Produces blurring, unnatural textures, or broken anatomy, especially in low-resolution variants or edge-cases. See Table 13 and Appendix M.4. These patterns highlight areas for further improvement.

# 7 CONCLUSION

We present SPADE, a large-scale multimodal dataset comprising 2,500 real-world toxic images, each annotated with three progressively detoxified variants and corresponding textual stories. Our dataset enables structured supervision for controllable toxicity reduction in image generation while preserving semantic fidelity. We present baseline results using a sequentially fine-tuned ControlNet, highlighting both the promise and challenges of image detoxification. Our work establishes a benchmark for future research in robust, context-aware, and safety-aligned multimodal content moderation.

## 8 ETHICAL STATEMENT

Our work presents a dataset creation and baseline models for detoxification in text-to-image generative models, with a focus on preserving semantic fidelity while mitigating harmful or sensitive content. All data used for training and evaluation were collected from publicly available, licensed websites, and no private or personally identifiable information was included.

When generating or evaluating potentially toxic content, we followed strict safety protocols and filtered outputs to avoid reinforcing stereotypes, misinformation, or graphic depictions of violence. Our evaluation framework emphasizes interpretability, reproducibility, and fairness, and we report both qualitative and quantitative metrics to highlight trade-offs in semantic preservation versus content detoxification. We acknowledge that detoxification may inadvertently suppress important sociopolitical narratives or cultural context, and we encourage future work to explore more nuanced, context-aware approaches.

This research aligns with responsible AI principles and has undergone internal review to ensure compliance with ethical standards in data handling, model usage, and human impact. All generated content was used solely for academic analysis, and no outputs were deployed in real-world applications.

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

# Overview of Appendices

- Appendix A: Detailed Related Work
- Appendix B: Further Details of SPADE Dataset Analysis
- Appendix C: GPT-4 Prompts
- Appendix D: Annotator Details

- Appendix E: Evaluation of Image Caption Quality
- Appendix F: Evaluation of contextual stories
- Appendix G: Examples of contextual stories
- Appendix H: Analyzing Detoxification Across Story Variants
- Appendix I: Evaluation of Toxic Image Variants
- Appendix J : Models Descriptions
- Appendix K: Hyper-parameters for reproducibility
- Appendix L: Variant-wise FID score
- Appendix M: Detailed Error Analysis

## A  DETAILED RELATED WORK

### A.1  IMAGE GENERATION

Several studies have been done in the field of image generation in the last decade across multiple paradigms, each focused on different forms of conditioning and control. These include basic image generation methods (Han et al., 2018; Cai et al., 2019; Zhu et al., 2018), image-to-image translation (Andreini et al., 2021; Mao et al., 2018; Lučić et al., 2019), sketch-to-image synthesis (Lu et al., 2018; Liu et al., 2017), conditional image generation (Jakab et al., 2018; Esser et al., 2018), few-shot image generation (Xu et al., 2019; Clouâtre & Demers, 2019), layout-based synthesis (Zhu et al., 2019; Zhao et al., 2019), pose-guided generation (Ma et al., 2018), and panoramic image creation (Zhang et al., 2019; Tripathi et al., 2019). Among these, T2I generation has emerged as a prominent subdomain. Early models relied on conditional GANs to produce images from textual prompts (Sharma et al., 2018; Xu et al., 2018), while recent diffusion-based approaches exhibit significantly improved fidelity, diversity, and controllability. These large-scale T2I models are trained on billions of image–text pairs scraped from the web, implicitly learning multimodal associations and often inheriting biases embedded in the training corpus. Models such as DALL·E (Ramesh et al., 2022), Imagen (Saharia et al., 2022), and Stable Diffusion (Rombach et al., 2022a) achieve impressive results by scaling both dataset size and model capacity. Among these, diffusion models have emerged as state-of-the-art due to their superior generative quality and controllability across diverse text prompts (Shetty & et al., 2024; Wimmer & Rebman, 2024). More recently, Sora (Liu et al., 2024), a text-to-video model developed by OpenAI, pushes the boundaries of generative modeling by simulating dynamic scenes with high visual fidelity and temporal coherence.

### A.2  CONTROLLABLE AND SAFE IMAGE GENERATION

While T2I generation models have achieved significant progress, they remain susceptible to generating harmful, biased, toxic, or non-ethical content, even when prompted with simpler language (Weidinger et al., 2022; Schramowski et al., 2022). This underscores the need for safer generation protocols that not only prevent harmful outputs but also preserve the integrity of the intended message (Schramowski et al., 2023a). Existing safety interventions typically rely on prompt filtering, red-teaming, or automated post-hoc moderation. Although these approaches are good, they often lack control over the degree and nature of detoxification, resulting in either over-correction or incomplete mitigation. Recent advancements in language-guided image editing, such as InstructPix2Pix (Brooks et al., 2023b) and Prompt-to-Prompt (Hertz et al., 2022), allow for local, semantically-aware modifications. Diffusion models offer improved controllability through various input modalities, including semantic layouts and conditioning labels (Nichol & Dhariwal, 2021; Ramesh et al., 2022). Some methods introduce explicit control mechanisms using auxiliary signals or guidance networks (Rombach et al., 2022b; Avrahami et al., 2023; Brooks et al., 2023b), while others opt for implicit strategies by refining the generation process of pre-trained models (Tumanyan et al., 2022; Mokady et al., 2022; Kwon & Ye, 2023; Kwon et al., 2023). Notably, Gafni et al. (2022) proposed a disentangled representation framework that enables controllable generation of coherent phrases with tunable ethical and stylistic attributes, highlighting the potential of latent-guided interventions in multimodal detoxification. However, their utility in ethical fine-tuning, where the goal is to reduce toxicity while retaining visual semantics, remains largely unexplored.

## A.3  IMAGE DETOXIFICATION AND SEMANTIC EDITING

Furthermore, while methods, such as InstructPix2Pix (Brooks et al., 2023b) and Prompt-to-Prompt (Hertz et al., 2022) enable text-guided image modifications, their primary focus lies in stylistic or aesthetic changes, rather than controlled ethical image generation. In continuation of the editing frameworks, there are computationally intensive fine-tuning techniques for each source image or subject (Kawar et al., 2023; Valevski et al., 2023), limiting scalability and applicability to real-time moderation. Efforts in text-based inpainting and semantic fusion (Zhang et al., 2020; Stap et al., 2020; Andonian et al., 2023; Avrahami et al., 2022) illustrate the potential of language-aligned visual editing, while techniques like textual inversion (Gal et al., 2022) and DreamBooth (Ruiz et al., 2023) enhance personalization and fidelity. However, most of these approaches prioritize expressiveness over ethical refinement, and are either costly to generalize across large datasets or susceptible to circumvention when deployed in unsafe settings (Schramowski et al., 2023b).

Despite such research, there exists a gap for a **controlled image-to-image detoxification with explicit ethical alignment** that remains largely lacking. Existing models lack mechanisms for progressive toxicity reduction in line with real-world context preservation and semantic consistency, crucial for socially responsible generative systems. To bridge this gap, we propose a novel **Image Detoxification and Semantic Editing** task and dataset that enables *step-wise toxicity reduction* of harmful images guided by multimodal conditioning (image + text). Our framework supports the development of safe and controllable generation pipelines, facilitating fine-grained moderation aligned with societal norms. Its deployment will empower platforms to automatically adapt visual content to user safety preferences, thereby reducing exposure to harmful imagery and enhancing digital well-being at scale.

# B  FURTHER DETAILS OF SPADE DATASET ANALYSIS

## B.1  HARMFUL KEYWORDS

| Categories | Keywords |
|---|---|
| Sexual | Woman in chains, Girl behind bars, Drugged prostitute with syringe, Bruised woman in brothel, Beaten escort with scars, Trafficked girl with rope marks, Shackled streetwalker, Abused woman in slum, Victimized sex worker with torn clothes, Underage prostitute with bruises, Forced prostitute holding drugs, Woman tied with rope, Beaten woman with whip marks, Helpless woman in dark alley, Exploited girl holding broken heels, Drugged escort with needle marks, Street prostitute with visible wounds, Woman in shackles in brothel, Trafficked girl with rope burns, Bruised call girl with visible cuts, Enslaved woman with iron chains, Beaten prostitute holding torn bag, Helpless woman behind a broken window, Abused girl clutching dirty pillow |
| Hate | Fat-shamed girl holding fast food, Body-shamed boy with scars, Obese person sitting on a broken chair, Thin girl with visible bones, Fat person with oversized clothes, Skinny person holding a tiny belt, |
| Violence | Terrorist and Violent Extremism, Terrorist holding an explosive device, Radicalized individual in extremist attire, Bomb explosion site with injured civilians, Militants attacking a civilian area, Hostage situation with captors and victims, Extremist propaganda materials displayed by individuals, Violence against civilians during an attack, Armed insurgents confronting local population, Suicide bomber preparing for an attack, Terrorist group gathering in secret location, Destruction from terrorism affecting families, Radical ideology representation through protests, Armed conflict zone with civilians caught in crossfire, Violence against law enforcement officers by extremists, Extremist rally or protest involving aggressive behavior, Civilian casualties from terrorist attacks depicted vividly, Explosive device preparation by militants, Militants targeting civilians during unrest, Terrorist group planning an attack, Violent uprising impacting local communities, Stalking Eating Disorders, Stalker watching from behind a tree, Surveillance camera capturing stalking behavior |
| Self-Harm | Visual of a group of students bullying a classmate, Scene of a student being pushed in a school corridor, Picture of a student isolated by peers in classroom, Image of a student being verbally attacked by classmates, Photo of a bullied student sitting alone at lunch, Visual representation of emotional bullying among students, Stalking Eating Disorders, Stalker watching from behind a tree, Surveillance camera capturing stalking behavior, Obsessive individual looking at photos of their target, Distressing weight loss imagery of a person, Anorexic individual looking at themselves in the mirror, Bulimic aftermath scene showing food waste, Emotional distress on the face of someone with an eating disorder, Victim of stalking looking over their shoulder anxiously, Hidden camera footage depicting stalking behavior, Fearful expression of someone being stalked |

Table 5: Harmful Keywords used to collect images

## B.2  T-SNE ANALYSIS OF SPADE

To qualitatively analyze semantic behavior across detoxification stages, we examine the t-SNE (van der Maaten & Hinton, 2008) projection of CLIP embeddings for a representative base toxic sample of 100 images, and their three detoxified variants. As shown in Figure 5, the original toxic samples (●) serve as the semantic anchor, typically positioned at the periphery of the embedding

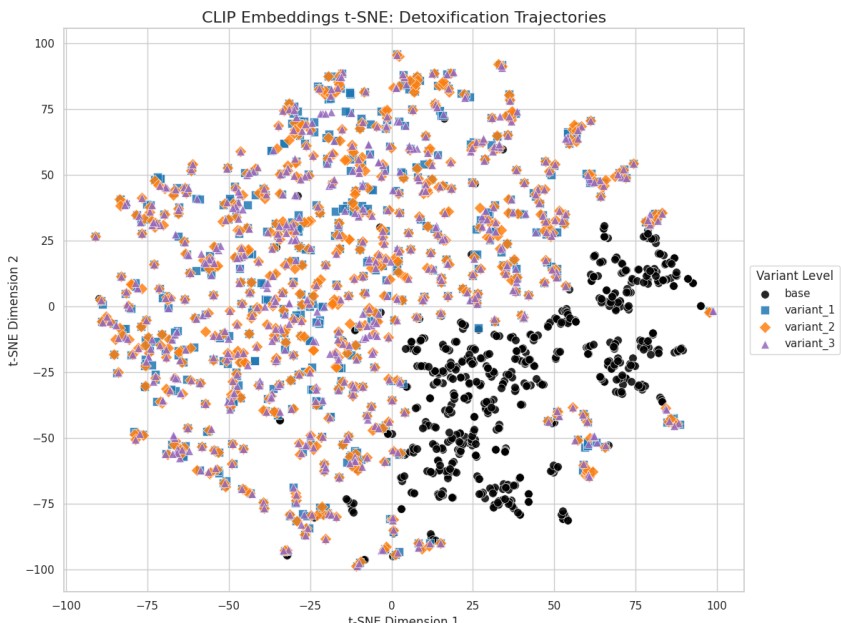

Figure 5: t-SNE visualization of CLIP image embeddings for three toxicity variants of 100 samples.

space due to their heightened toxicity and distinct features. As detoxification begins, variant $\mathcal{V}_1$ (■) initiates a directional shift away from the base, reflecting early-stage mitigation of toxic traits while retaining core semantics. Variant $\mathcal{V}_2$ (◆) continues this trajectory, moving further from the base and closer to the detoxified cluster. It demonstrates more pronounced semantic refinement, often involving structural rephrasing or contextual rebalancing. Finally, variant $\mathcal{V}_3$ (▲) exhibits the most compact clustering, indicating convergence toward a detoxified yet semantically coherent representation. The smooth progression highlights effective toxicity reduction while preserving semantic integrity.

## C  GPT-4 PROMPTS

### C.1  DETOXIFIED CAPTION VARIANT GENERATION

*You are an AI detoxifier. Your role is to detoxify the given image in such a way that it changes the image caption by decreasing the toxicity of the image caption.*
*If the toxic image caption is "A person with a gun in his hand." Your response should be decreasing the toxicity with three levels: 1. "A person with a stick in his hand." 2. "A person with a stone in his hand" 3. "A person with a toy in his hand".*
*If the toxic image caption is "A man is being kicked by another man in a field." Your response should be decreasing the toxicity with three levels: 1. "A man is arguing with another man in a field." 2. "A soccer ball is being kicked by a man in a field." 3. "A baby is playing with another man in a field."*

### C.2  CONTEXTUAL STORY CONSTRUCTION

*Tell me a story $\mathcal{S}^k$ in which the given toxic image fits in. The story should be no more than 10 sentences. Of these, at least 4 sentences should logically occur before the image caption, and at least 4 sentences should logically occur after the image caption. Please also include the above caption of the image in the story in a natural way.*

## C.3 KNOWLEDGE RELEVANCE (KR)

**Role:** You are an expert evaluator tasked with assessing whether a detoxified image remains factually accurate and contextually aligned with real-world knowledge after toxic elements have been removed or replaced.

**Input:** A pair of images: *(i)* the original reference image, which may contain toxic content but is grounded in real-world knowledge, and *(ii)* the detoxified image, where toxic elements have been replaced with more neutral or benign counterparts.

**Task:** Your task is to assess the knowledge relevance of the detoxified image, i.e., whether it maintains accuracy and consistency with the real-world knowledge conveyed in the reference image. Specifically, evaluate whether the core informational content remains valid after detoxification.

**Scoring Scheme (5-point Likert Scale):**

- **1:** *Completely Irrelevant:* The image introduces clear factual errors or misrepresents real-world knowledge significantly.

- **2:** *Mostly Irrelevant:* Major facts are incorrect, though some minor elements might be accurate.

- **3:** *Somewhat Relevant:* A mix of accurate and inaccurate information; key concepts may be preserved but distorted.

- **4:** *Mostly Relevant:* Largely factually correct with only minor inconsistencies.

- **5:** *Completely Relevant:* Fully consistent with real-world facts; all key information remains intact and accurate.

**Guidelines:**

- Focus strictly on factual and knowledge-level correctness, not aesthetics or style.

- Consider whether key entities, actions, settings, or concepts from the reference remain intact.

- Ignore surface-level features unless they contribute to factual meaning.

**Output:** {*Score: 1 to 5, Justification: Brief explanation citing what knowledge is preserved or lost.*}

## C.4 CONTEXT PRESERVATION (CP)

**Role:** You are an AI expert evaluator tasked with determining whether the contextual integrity of a detoxified image is preserved. This includes assessing whether the original meaning, communicative intent, emotional tone, and narrative elements are maintained after removing or replacing toxic content.

**Input:** A pair of images: *(i)* the original reference image, which may contain toxic elements, and *(ii)* the detoxified image, where such elements have been modified.

**Task:** Evaluate the extent to which the detoxified image preserves the original image's communicative purpose. Consider the continuity of message, implied tone, emotional implication, humor, and narrative structure.

**Scoring Scheme (5-point Likert Scale):**

- **1:** *Completely Lost:* The detoxified image conveys an entirely different or contradictory message.
- **2:** *Mostly Lost:* Major shifts in message or tone; minimal thematic overlap.
- **3:** *Somewhat Preserved:* Partial retention of meaning; key contextual elements are weakened or altered.
- **4:** *Mostly Preserved:* Core intent and message largely intact with minimal semantic drift.
- **5:** *Completely Preserved:* Full retention of original meaning, tone, and communicative intent.

**Guidelines:**

- Focus on whether the detoxified image communicates the same message as the original.
- Evaluate emotional tone, humor, rhetorical force, and narrative elements.
- Ignore artistic or stylistic differences unless they affect interpretability.

**Output:**{*Score:1 to 5, Justification: Brief explanation citing which aspects of meaning or intent were preserved or altered*}

## D ANNOTATOR DETAILS

To ensure the reliability and depth of the qualitative evaluation of the generated stories and image captions, we involved two independent annotators with relevant domain expertise. One annotator was a final-year Ph.D. scholar specializing in multimodal natural language processing and toxic content understanding, while the second annotator was an M.Tech. student with academic training in computer vision and ethical AI. Both annotators had prior experience in dataset curation and annotation schemes for NLP and vision-language alignment tasks. Prior to the annotation task, they were briefed with detailed guidelines on rating story quality based on narrative coherence, emotional framing, and toxicity reduction. Disagreements were resolved through discussion sessions, and consensus ratings were used for final scoring and analysis.

## E EVALUATION OF IMAGE CAPTION QUALITY

In this section, we evaluated the image caption quality generated by GPT-4. In Figure 6, we have shown a few examples from our dataset. We analysed 100 such images and annotated the caption quality manually on a scale of 1 to 5. A significant proportion of the captions, specifically, those rated 5, demonstrated excellent alignment between visual content and linguistic description. These high-quality captions exhibited accurate representation of subject identity (e.g., gender, age), emotional cues (e.g., anger, frustration, distress), and spatial or interactional context (e.g., arguments in office or street environments). For instance, captions such as "A student sitting on the floor with her head in her hands, looking stressed" or "Children bullying another child in the library" provided detailed and affect-rich interpretations that go beyond mere object detection. They captured the underlying social dynamics and moral valence, which are critical for downstream tasks like narrative generation, causal explanation, and content moderation. Such samples validate the capa-

| Reference Image | Caption | Score | Justification |
|---|---|---|---|
| | A drone being prepared for a mission in a hangar. | 5 | Clearly captured the image information. |
| | A man standing in a room, wearing a mask and showing his belly. | 5 | Clearly captured the image information. |
| | A person shielding their face with their hands. | 4 | Clearly captured the image information. |
| | A military drone in a hangar. | 4 | The spatial location (hangar) is not clearly indicated in the image context. |
| | A group of people having a discussion in a meeting room. | 4 | The scene does not appear to take place in a typical meeting room, reducing contextual precision. |
| | A person standing behind a shattered mirror. | 4 | The subject appears to be a woman; the caption fails to specify gender, which impacts narrative precision. |
| | A person is being pointed at by multiple hands. | 3 | The caption lacks gender identification and fails to convey the negative emotional context, such as shame or fear. |
| | A group of children standing in a school hallway, one child holding their head. | 3 | The image likely depicts bullying, but emotional cues like sadness or fear are missing, and finger-pointing is not described. |
| | A person curled up in a fetal position, suspended in mid-air. | 3 | The figure depicted does not resemble a human person but possibly a creature or abstraction; the caption misrepresents this. |
| | A person throwing a Molotov cocktail by an empty swimming pool. | 2 | The image actually shows the Molotov being thrown toward a stadium amid a riot; the caption misrepresents the scene and downplays its severity. |

Table 6: Image caption example sample from our dataset

bility of GPT-4 in generating contextually nuanced and morally salient captions, which can serve as strong supervisory signals for fine-tuning multimodal reasoning systems. However, we also identified systematic limitations in the low to mid-scoring captions ($scores \leq 4$). A few captions lacked gender specificity, often using the neutral term "person" despite the presence of identifiable gender cues. This reduces narrative fidelity in applications sensitive to gender representation. In several other cases, emotion recognition was either missing or understated, especially in scenarios involving bullying, confrontation, or psychological distress, e.g., "A child standing in a library with other children in the background" failed to capture the visible fear and exclusion implied in the image. Furthermore, some captions presented semantic inaccuracies such as incorrect object identification or scene misinterpretation, for example, describing a stadium-based fire scene as an "empty swimming pool." These issues highlight the need for incorporating caption justification layers, where annotators or models assess the presence of key semantic components, enabling fine-grained error correction and targeted improvements in vision-language alignment.

| Image | Story |
|---|---|
|  | John had always been fascinated by technology and its potential to change the world. He joined the Air Force to work on cutting-edge projects, and his latest assignment was no exception. The team had been working tirelessly for months, developing a new type of drone. The day had finally come to test their creation. As John and his colleague walked into the hangar, they saw the drone being prepared for its first mission. They double-checked all the systems and ensured everything was in place. With a final thumbs-up, the drone took off into the sky. The team celebrated their success, knowing this was only the beginning. |
|  | John and Emily had been friends for years, sharing countless memories. Recently, their friendship was strained by a misunderstanding. They decided to meet in person to clear the air. As emotions escalated, they faced each other with fists raised in anger. Just as things were about to get out of hand, Emily lowered her fists and urged calm discussion. John followed her lead, and together they talked through the issue. By the end of the conversation, they apologized and promised to rebuild their friendship stronger than before. |
|  | John had always been an adventurous soul, often exploring the woods and hiking trails near his home. One day, he decided to venture deeper into the forest than he ever had before. As he navigated through the dense foliage, he accidentally brushed against a patch of poisonous plants. Unaware of the danger, he continued his hike, enjoying the serenity of nature. By the time he returned home, John started feeling an intense itching sensation on his skin. The next morning, he woke up to find his torso and arms covered in red, inflamed spots. A person with a severe skin condition on their torso and arms. Alarmed, he quickly made an appointment with his doctor. The doctor diagnosed him with a severe allergic reaction to the poisonous plants he had encountered. John was prescribed medication to alleviate the itching and inflammation. Over the next few days, he diligently followed the doctor's advice, applying ointments and taking antihistamines. Gradually, the redness and swelling began to subside. John learned a valuable lesson about the importance of being cautious in unfamiliar environments. |
|  | John and Mike had been best friends since childhood, but recently, their friendship had been strained due to a misunderstanding. They decided to meet at the park to talk things out. As they began discussing their issues, emotions ran high, and their voices grew louder. The tension escalated quickly, leading to a heated argument between two men with clenched fists. Passersby started to notice the commotion and gathered around, concerned about the situation. Just as it seemed like things might get out of hand, a mutual friend, Sarah, arrived and intervened. She managed to calm both men down and encouraged them to talk things through more peacefully. John and Mike realized how much their friendship meant to them and decided to resolve their differences. They apologized to each other and hugged, promising to communicate better in the future. The crowd dispersed, relieved that the situation had been peacefully resolved. |
|  | In the bustling city of New York, tensions had been rising for weeks. The economic downturn had left many people frustrated and desperate. John, a young man trying to make ends meet, found himself in the middle of a heated argument with a group of strangers. The argument quickly escalated, drawing the attention of passersby. A group of people are fighting on the street. The scene was chaotic, with shouts and scuffles breaking out. Police sirens could be heard in the distance, signaling that help was on the way. As the authorities arrived, they managed to break up the fight and calm the situation. John, bruised but relieved, was taken aside to give his statement. The crowd slowly dispersed, leaving the street eerily quiet. In the aftermath, the community came together to address the underlying issues that had led to the confrontation, hoping to prevent such incidents in the future. |
|  | Once upon a time, there was a group of friends who loved spending time together at school. They would meet every morning in the hallway, sharing stories and laughing. One day, they were excitedly discussing their plans for the upcoming weekend. Suddenly, they heard a loud noise from the other end of the hallway. A group of children standing in a school hallway, one child holding their head. The noise startled them, and they turned to see what had happened. One of the children had accidentally bumped into a locker and was holding their head in pain. The friends quickly gathered around, concerned for their friend. They helped the child to the nurse's office to make sure everything was okay. After a few minutes, the child felt better and thanked their friends for their support. They all decided to be more careful in the future and continued their day with smiles on their faces. The incident brought them even closer together, reminding them of the importance of looking out for one another. |
|  | Once upon a time, there was a group of friends who loved spending time together at school. They would meet every morning in the hallway, sharing stories and laughing. One day, they were excitedly discussing their plans for the upcoming weekend. Suddenly, they heard a loud noise from the other end of the hallway. A group of children standing in a school hallway, one child holding their head. The noise startled them, and they turned to see what had happened. One of the children had accidentally bumped into a locker and was holding their head in pain. The friends quickly gathered around, concerned for their friend. They helped the child to the nurse's office to make sure everything was okay.— After a few minutes, the child felt better and thanked their friends for their support. They all decided to be more careful in the future and continued their day with smiles on their faces. The incident brought them even closer together, reminding them of the importance of looking out for one another. |

Table 7: Examples of GPT-4 Generated Stories for Detoxification Evaluation

## F  EVALUATION OF CONTEXTUAL STORIES

In this section, we analyze the effectiveness of the contextual stories generated using GPT-4 for detoxifying visual content. Figure 7 illustrates a representative sample from our SPADE dataset, and we manually evaluated 100 such instances on a 1–5 quality scale, assessing spatial relevance, emotional plausibility, and narrative coherence.

Overall, the GPT-4 model demonstrates strong capability in transforming static toxic images into plausible, emotionally grounded narratives. Specifically, it excels in setting scenes with spatial and temporal precision. For example, in the first example sample ("A drone in a hangar"), the generated story begins with an airman preparing for a mission, creating a logical and semantically rich lead-in. Emotional and causal framing is another significant strength: in the second example ("A man and a woman with raised fists"), the narrative presents the characters as old friends whose conversation escalates into a disagreement, thus reframing the visual aggression in humanizing terms. The required "before-and-after" narrative structure ensures a progression from tension to resolution; the fourth example ("Two men with clenched fists") exemplifies this by introducing a mutual friend post-conflict to mediate reconciliation. Moreover, in visually grounded but non-emotional cases, such as the third example sample ("Person with a skin condition"), the model introduces a medically plausible backstory (contact with toxic plants), converting a disturbing image into a health-focused narrative. Collectively, these results affirm GPT-4's efficacy in semantic preservation and soft image detoxification through narrative reframing.

**Challenges and Failure Cases.**  Despite these strengths, several limitations emerge. First, emotional ambiguity remains a concern. For instances such as example sample fourth or sixth in Table 7, captions depicting hand gestures or distressed children lack explicit affective descriptors (e.g., "sad," "anxious"), reducing interpretive clarity. Second, narrative misinterpretation of action is evident in borderline ambiguous cases. For instance, the fifth example sample ("A group of people fighting on the street") suffers from unclear disambiguation, whether it is a protest or a brawl, potentially undermining the detoxification goal. Third, the generated stories occasionally exhibit generic protagonist modeling, using placeholder names like "John" or omitting relational roles (e.g., "his colleague" vs. "another person" in example sample 1), limiting empathy. More nuanced restorative endings (e.g., reflective dialogue, peer mediation) are infrequent but desirable for advancing ethically grounded narrative detoxification.

## G  EXAMPLES OF CONTEXTUAL STORIES

Here, we present the story for the reference toxic image illustrated in Fig. 2 and Fig. 3, following our story-guided detoxification framework.

*Late at night, Alex found herself exploring the basement of an old theater that had been closed for decades. As she stepped cautiously into the narrow corridor, dim lights flickered overhead, revealing ropes once used for stage effects swaying gently from the ceiling. She paused for a moment, taking in the creepy silence and the scent of aged wood and dust. Despite the unsettling atmosphere, her curiosity about the theater's history urged her onward. A woman standing alone in a dark, narrow corridor with shadowy walls, and ropes and tangled wires hanging around, she felt like part of a forgotten story waiting to be rediscovered.*

*John and Mike had been friends for years, but recently their relationship had become strained. They had a disagreement over a business deal that went sour, and tensions were high. One day, they decided to meet at the gym to try to resolve their differences. However, the conversation quickly escalated into a heated argument. A person is attacking another person with a knife. John, in a fit of rage, grabbed a knife from his bag and lunged at Mike. Mike tried to defend himself, but the situation was getting out of control. Other gym-goers quickly intervened and managed to separate the two men. After the incident, both John and Mike were taken to the police station for questioning. They realized the gravity of their actions and regretted letting their emotions get the best of them. They decided to seek professional help to manage their anger and work on their communication skills. Over time, they were able to rebuild their friendship and move past the incident.*

Table 7 shows more examples of stories generated by GPT-4.

## H  ANALYZING DETOXIFICATION ACROSS STORY VARIANTS

To evaluate the effectiveness of our detoxification approach, we analyze the evolution of story representations across three progressively de-toxicified image variants (Variant 1, Variant 2, and Variant 3) for each toxic image in SPADE. This assessment focuses on three core dimensions: narrative consistency, toxicity content drift, and emotional polarity shift.

### H.1  STORY CONSISTENCY ACROSS VARIANTS

We observe that the average character length of stories remains stable across the three variants: 1524.7 (Variant 1), 1522.1 (Variant 2), and 1520.2 (Variant 3). This similar length indicates that while the toxicity is reduced, the semantic richness and syntactic completeness of the story are preserved. For instance, in toxic_105.png (a person throwing a Molotov cocktail by a stadium), all three variants maintain the same core structure by depicting protest-related imagery, but the details and outcomes are progressively detoxified while preserving the same scene actors, location, and temporal progression. This demonstrates that the model does not simplify or abstract the scenario at the cost of narrative loss.

### H.2  TOXICITY KEYWORD REDUCTION AND CONTENT DRIFT

We qualitatively observe a controlled reduction of toxic cues across variants, such as aggression, physical violence, or morally loaded actions. Lexical substitutions are employed to de-escalate the visual implication without altering the scene composition. For example, in toxic_1020.png (a man yelling at another person), Variant 1 starts with a heated confrontation involving loud accusations. In Variant 2, this is reframed as a tense professional disagreement. By Variant 3, the story becomes a workplace miscommunication resolved by a team leader. Violent tone is replaced with discourse driven tension, yet characters and setting remain consistent.

### H.3  EMOTIONAL TRAJECTORY AND POLARITY SHIFT

The most distinctive aspect of detoxification is the reframing of emotions. The initial variant often introduces sharp, negative emotion elements like anger, fear, aggression etc., while successive variants reconstruct these emotions with relatively positive dimensions. For example, in toxic_1019.png (two men with clenched fists), the Variant 1 story describes an impending physical altercation rooted in betrayal. Variant 2 identifies a misunderstanding involving a third party. Variant 3 resolves this misunderstanding through empathetic dialogue facilitated by a mutual friend. This exemplifies the emotional polarity shift toward harmony without removing the original tension anchor.

## I  EVALUATION OF TOXIC IMAGE VARIANTS

To assess the effectiveness and semantic fidelity of our detoxified image dataset, we present a qualitative comparison of three detoxified variants ($\mathcal{V}_1$, $\mathcal{V}_2$, $\mathcal{V}_3$) for each toxic image sample, as shown in Table 9. Across most examples, the generated variants demonstrate a systematic reduction in visual toxicity while maintaining semantic alignment with the original context. For instance, the transition from a disturbing image of a deceased animal to its detoxified counterparts shows a meaningful semantic shift towards peaceful barnyard scenes, effectively eliminating the distressing cues without losing core thematic elements (i.e., presence of animals, environment). Similarly, toxic content depicting drug usage is successfully transformed into constructive academic or creative scenes, evidencing the model's ability to reinterpret intent while ensuring detoxification. However, in some instances, for example, the fourth row, we observe that variant $\mathcal{V}_3$ exhibits a cartoonish rendering that diverges significantly from the photorealistic nature of the original toxic sample. While the semantic theme (i.e., interaction with an elephant) is loosely preserved, the realism and context fidelity are compromised due to style drift. Such samples highlight the trade-off between detoxification strength and content preservation at deeper variant stages, warranting further refinement in variant

| Toxic Image | Story Consistency | Toxicity Shift (Lexical/Substantive) | Emotional Polarity Shift |
|---|---|---|---|
|  | All variants depict a protester throwing an object in a public setting, maintaining consistent actors and location. | Variant 1 uses "Molotov cocktail"; Variant 2 uses "glass bottle"; Variant 3 refers to "symbolic act of protest". | Ends with violence in Variant 1; de-escalation in Variant 2; peaceful protest resolution in Variant 3. |
|  | Scene remains at a workplace involving two male characters across all variants. | Variant 1: "yelling at colleague"; Variant 2: "arguing over a mistake"; Variant 3: "clarifying disagreement with manager". | Emotion shifts from anger → stress → mutual understanding. |
|  | Consistent framing of two friends in conflict in all three variants. | "Fists clenched in rage" → "argument turns tense" → "friend mediates resolution". | Emotions move from violent escalation → frustration → reconciliation. |
|  | School hallway and student characters are retained across versions. | Bullying is reframed: Variant 1 shows ridicule; Variant 2 implies teasing; Variant 3 highlights concern from peers. | Trajectory: social exclusion → ambiguity → emotional support. |
|  | An individual's medical condition remains central across variants. | Variant 1 mentions "severe rash"; Variant 2: "allergic reaction"; Variant 3: "skin irritation from plant contact". | Emotionally shifts from alarm → worry → resolved curiosity. |

Table 8: Illustrative examples of story detoxification across three variants

generation strategies to balance both. Overall, the dataset offers strong visual detoxification quality with increasing abstraction, making it a useful resource for benchmarking gradual toxic-to-safe transitions.

## J  MODELS DESCRIPTIONS

- **Stable Diffusion (Zero-shot)**: This baseline employs the original Stable Diffusion model without any task-specific fine-tuning. It generates outputs solely based on its pre-trained capabilities and lacks awareness of toxicity or detoxification objectives.

- **ControlNet (Zero-shot)**: The ControlNet model is used in its pre-trained form without any adaptation to the detoxification task. It leverages structural control (e.g., pose, edges) for generation, but lacks learned alignment with detoxification prompts.

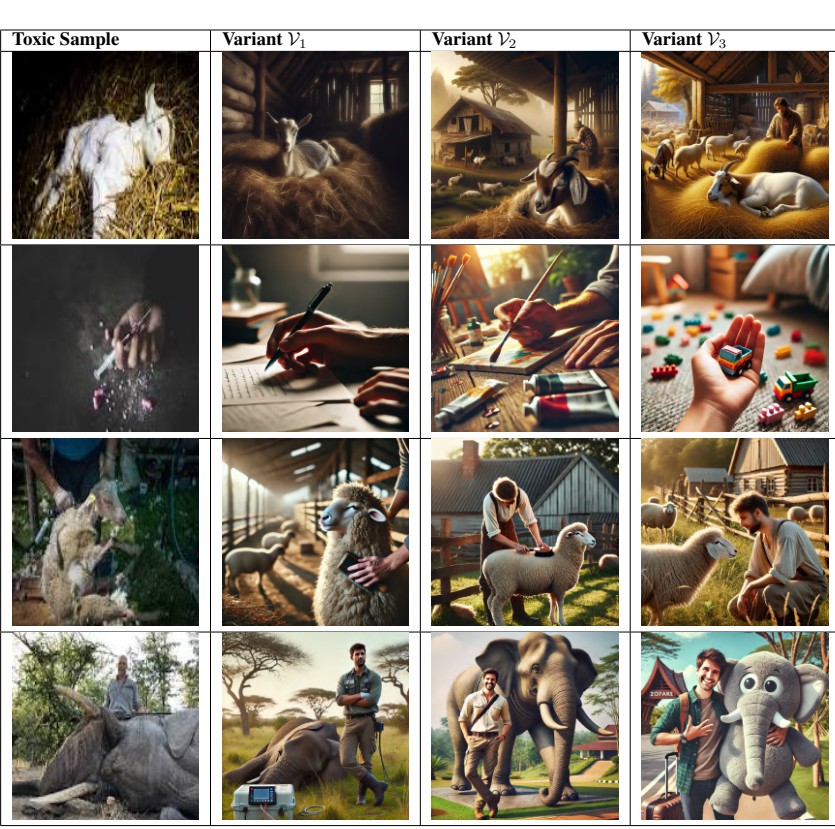

Table 9: Qualitative comparison of detoxified image variants ($\mathcal{V}_1$, $\mathcal{V}_2$, $\mathcal{V}_3$) with respect to the original toxic sample.

| Model | Variants-wise FID ↓ | | | | | |
|---|---|---|---|---|---|---|
| | $\mathcal{V}_1$-$\hat{\mathcal{V}}_1$ | $\mathcal{V}_2$-$\hat{\mathcal{V}}_2$ | $\mathcal{V}_3$-$\hat{\mathcal{V}}_3$ | $\mathcal{I}$-$\hat{\mathcal{V}}_1$ | $\mathcal{I}$-$\hat{\mathcal{V}}_2$ | $\mathcal{I}$-$\hat{\mathcal{V}}_3$ |
| Stable Diffusion (Zero-Shot) | 215.7381 | 218.1704 | 209.7249 | 231.28 | 261.28 | 232.38 |
| Stable Diffusion (Finetuned) | 242.2874 | - | - | 271.3754 | - | - |
| Safe Diffusion | 190.0590 | 242.1599 | 151.1310 | 218.85 | 197.643 | 241.864 |
| ControlNet (Zero-Shot) | 196.4298 | 202.3811 | 202.9864 | 226.386 | 241.386 | 257.387 |
| ControlNet (Finetuned) | 152.4207 | 153.6061 | 151.1990 | 205.3857 | 221.3822 | 219.3864 |
| Sequential ControlNet (Proposed) | 151.5077 | 152.5274 | 226.2852 | 126.19 | 172.28 | 202.28 |

Table 10: Variants-wise FID scores across detoxification models. Lower values indicate better perceptual fidelity between original and detoxified images.

- **Stable Diffusion (Fine-tuned)**: This model represents the first baseline where we fine-tuned the original Stable Diffusion architecture using our SPADE dataset. During training, only the story corresponding to each toxic image was used as the conditioning text prompt. The model was optimized to generate a single detoxified image per input, aiming to semantically align with the story while reducing visual toxicity.

- **ControlNet (Fine-tuned)**: In this baseline, we fine-tune the ControlNet model on our SPADE using both the toxic image and its corresponding story prompt as conditioning inputs. Unlike the previous Stable Diffusion variant, which relied solely on textual conditioning, this setup integrates visual guidance from the toxic image to better preserve structural fidelity during detoxification. Although the generated detoxified variants exhibit improved alignment and control, they are produced independently and do not form a semantically progressive sequence.

- **Safe Diffusion (Schramowski et al., 2023b)**: This model (AIML-TUDA/stable-diffusion-safe) generates multiple detoxified variants for a given toxic image by leveraging a safety-aware diffusion architecture. Unlike ControlNet-based approaches, it does not utilize the original image as a conditioning input and operates purely on the detoxification story prompt. While the model is capable of producing three detoxified samples per toxic instance, the variants are generated independently and lack a semantically coherent progression. This model also does not execute sequential consistency across variants and does not retain structural fidelity to the original image.

- **Sequential ControlNet (Proposed)**: Our proposed model introduces a sequential multi-stage generation pipeline that progressively detoxifies the image using intermediate variants. It conditions on both the image and the associated story, enabling finer semantic alignment, preservation of contextual cues, and reduction of hallucinations.

## K HYPER-PARAMETERS FOR REPRODUCIBILITY

To ensure the reproducibility of our ControlNet fine-tuning experiments, we provide a comprehensive list of hyperparameters and training settings used throughout our pipeline. These include model initialization, dataset specifications, optimization strategies, and hardware configurations. Table 14 outlines the exact values employed during training and evaluation.

## L VARIANT-WISE FID SCORE

The variant-wise FID score analysis also reveals that our Sequential ControlNet (Proposed) model consistently outperforms all baseline models, including Stable Diffusion (Zero-shot), Stable Diffusion (Fine-tuned), Safe Diffusion (Schramowski et al., 2023b), ControlNet (Zero-shot), and ControlNet (Fine-tuned), in preserving perceptual realism across detoxified variants. It achieves the lowest FID when comparing the original toxic image $\mathcal{I}$ to detoxified outputs (e.g., $\mathcal{I}$-$\hat{\mathcal{V}}_1$ = 126.19), indicating strong visual alignment with minimal degradation. While FID for $\mathcal{V}_3$-$\hat{\mathcal{V}}_3$ is slightly higher (226.28) due to larger semantic deviation in the detoxification trajectory, overall scores across all variants highlight the effectiveness of our progressive, image-conditioned generation strategy in maintaining fidelity and structure throughout the detoxification pipeline.

## M  DETAILED ERROR ANALYSIS

### M.1  ERRORS DUE TO HALLUCINATION

In our proposed sequential ControlNet pipeline, despite improvements in semantic preservation and detoxification efficacy, we observe a few instances of hallucinated visual elements that are not grounded in the original image or the guiding story prompt. For example, in Example 1 in Figure 6, the reference image depicts a child holding a bowl. While Variant 1 and Variant 2 maintain structural fidelity, the hallucinated bowl pattern in Variant 2 introduces stylized ornamentation absent in the original. In Variant 3, although detoxified, the child is shown drinking directly from a large white bowl, an exaggerated abstraction of the original object, resulting in a semantically misleading portrayal. Similarly, Example 2 begins with a toxic input showing a blood-like smear on the wall near a bed. As detoxification progresses, the hallucination deteriorates: Variant 1 generates a misaligned color smear and an over-decorated room with inconsistent shadows, while Variant 3 introduces an entirely unrelated black insect over a grid background, completely disconnected from the original context. These samples highlight the limitations of our proposed model in handling complex detoxification prompts while maintaining spatial and semantic alignment, especially under abstract or ambiguous story conditions. This necessitates future work in grounding control signals more robustly and integrating stronger alignment losses to mitigate semantic hallucination.

### M.2  ERRORS DUE TO CONTEXTUAL DRIFT

Despite leveraging conditioning from both the toxic input and the associated detoxification stories, our proposed sequential model occasionally exhibits semantic inconsistencies that result in contextual drift. We observe this error in examples such as those illustrated in Table 11.

In the first row, the original toxic image clearly depicts a school-based bullying incident, featuring a distressed child seated in a classroom with aggressive students in the background. The associated detoxified variant generated by our model, however, replaces the indoor academic environment with a neutral outdoor park bench scene. Although the generated image maintains an overall human-centered composition, the important context of institutional bullying is entirely lost. This drift from an educational context to a generic outdoor setting weakens the story grounding and disrupts the intended semantic alignment with the input story. In the second example, the toxic image illustrates a protest scene, characterized by a prominent protester and background crowd, central visual elements that signify socio-political stress. In contrast, the detoxified image fails to retain these elements: the protester is omitted and the crowd is absent, resulting in a depopulated urban scene devoid of its activist framing. This transformation significantly alters the narrative trajectory, stripping the image of its political relevance and emotional impact. These examples demonstrate that while our sequential model can perform content-level detoxification effectively, it sometimes fails to preserve scene-level semantic structures. Such cases emphasize the need for incorporating stronger alignment mechanisms to anchor the generation to both visual and contextual priors more reliably.

### M.3  ERRORS DUE TO INSTRUCTION MISINTERPRETATION

Our proposed model occasionally demonstrates partial or incorrect adherence to the detoxification story, particularly when the story involves nuanced semantic shifts or compound directives. Such failures often arise in cases where the intended meaning of the story lacks direct visual grounding or contains subtle linguistic cues. These instances suggest limitations in how well the model interprets and aligns with nuanced textual conditioning, particularly when the story is not directly grounded in visual cues or is semantically intricate in the original content. As illustrated in Table 12, this error category captures cases where the model misinterprets the affective tone or situational intent described in the detoxification prompt. In the first example, the toxic image portrays a visibly distressed girl, intended to be retained with an emotionally somber tone. However, the detoxified variant incorrectly depicts the girl as smiling, thereby misrepresenting the core affective state. Similarly, in the second example, the original image communicates a theme of exclusion and isolation, visually emphasized through a solitary walk in the rain. The detoxified output, however, depicts a cheerful boy engaged in play, effectively negating the narrative's emotional weight and rendering the transformation semantically incoherent. These examples indicate that the model may struggle with subtle emotional shifts or prompts that require grounding in less visually explicit cues, particularly

when detoxification involves preserving complex social or psychological contexts. Such misinterpretations undermine the fidelity of the detoxification process and highlight the need for improved visual-textual alignment mechanisms.

## M.4 ERRORS DUE TO VISUAL ARTIFACTS

Despite achieving semantic alignment with the detoxification prompt, our proposed model occasionally exhibits significant visual degradation in the form of rendering artifacts. As highlighted in Table 13, these artifacts include anatomical distortions (e.g., broken arms, warped limbs, and facial asymmetries), unnatural textures, and pervasive blurring, especially in structurally dense or low-resolution regions. For instance, in the first example, although the core scene remains preserved, the visual integrity is compromised by severely malformed arms and facial irregularities, diminishing the realism and interpretability of the generated image. In the second case, the detoxified output presents overly blurred textures and poorly rendered lockers, undermining the overall visual fidelity despite maintaining the intended content. These issues are symptomatic of limitations in the model's spatial coherence and sampling precision under complex or ambiguous visual contexts. While the semantic target is often achieved, such artifacts adversely affect downstream usability.

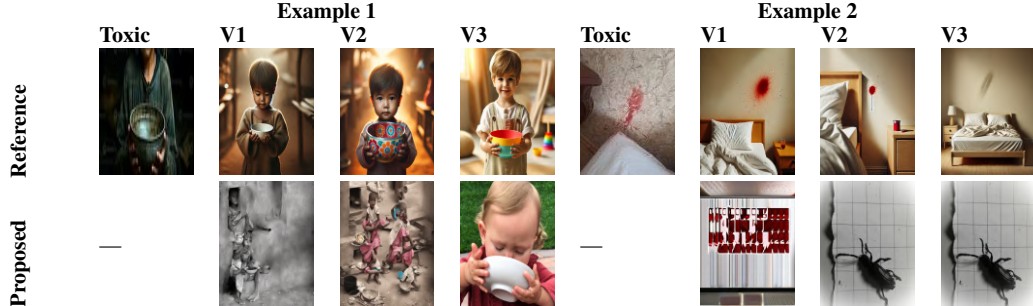

Figure 6: Qualitative Examples of Hallucination Errors Across Detoxification Stages

| Toxic Sample | Detoxified Variant | Observed Issue |
| --- | --- | --- |
| | | Subject and scene context altered, the classroom bullying scene becomes a generic park bench scene. |
| | | Protester disappears, crowd removed, loss of protest setting undermines political narrative. |

Table 11: Examples of Contextual Drift in detoxified outputs where core semantic elements are not preserved.

| Toxic Sample | Story | Detoxified Variant 1 | Observed Issue |
|---|---|---|---|
|  | Emily had always been a bright student, but after moving to a new city, she struggled to adjust to her larger school and missed her old friends. One difficult afternoon, she slipped away to a quiet corner and broke down in tears, overwhelmed by loneliness. Her teacher, Mrs. Thompson, found her sitting alone, visibly distressed. She gently sat beside Emily, listened patiently, and reassured her that these feelings were valid. With comforting words and a promise of support, Mrs. Thompson helped Emily feel a little less alone in her new journey. |  | Sad emotional tone misinterpreted, girl shown smiling rather than crying. |
|  | Bob stood quietly on the sidewalk as rain poured down around him. He had just left a tense meeting and needed time to process everything. Wrapped in his black rain jacket, hood pulled tight, he watched the cars pass by, lost in thought. The city felt distant, blurred by the downpour and his own uncertainty. |  | Exclusion theme misrepresented, lonely rainy walk replaced by cheerful playing boy. |

Table 12: Examples of **Instruction Misinterpretation** due to incorrect understanding of textual intent.

| Toxic Sample | Detoxified Variant | Observed Issue |
|---|---|---|
|  |  | Broken arms, distorted faces, severe rendering issues despite semantic alignment. |
|  |  | Blurry visuals and unnatural locker textures degrade image quality. |

Table 13: Examples of **Visual Artifacts** demonstrating rendering failures in generated detoxified images.

| Parameter | Value |
|---|---|
| Pretrained Base Model | *stabilityai/stable-diffusion-2-1-base* |
| Resolution | *512 × 512* |
| Batch Size | *1* |
| Gradient Accumulation Steps | *2* |
| Learning Rate | *1e-5* |
| Optimizer | *8-bit Adam* |
| Mixed Precision | *fp16* |
| Gradient Checkpointing | Enabled |
| Epochs | *5* |
| Warmup Steps | *0* |
| Checkpointing Steps | *50* |
| Dataloader Workers | *8* |
| TF32 Enabled | *True* |
| Validation Images | 2 reference images (custom) |
| Validation Prompt | Custom multimodal narrative prompt |
| Tokenizer | *CLIPTokenizer* (*openai/clip-vit-base-patch32*) |
| Evaluation Metrics | LPIPS, CLIPScore, SSIM, ERR Score, Human Ratings |
| Hardware | 4 × NVIDIA A100 GPUs |
| Framework | HuggingFace Accelerate |
| Logging | Weights & Biases (wandb) |
| Push to Hub | Enabled |

Table 14: Hyperparameters and Training Configuration for ControlNet Fine-Tuning

