# OpenReview forum: "SPADE: SEMANTIC-PRESERVING ADAPTIVE DETOXIFICATION OF IMAGES"
_ICLR.cc/2026/Conference — ICLR 2026 Conference Withdrawn Submission_

### Official Review · Reviewer_FLXg · 2025-10-14

**Soundness:** 4
**Presentation:** 3
**Contribution:** 4
**Rating:** 4
**Confidence:** 4

**Summary:**

This paper introduces SPADE, a novel dataset and benchmark for graded, semantic-preserving detoxification of harmful images. Unlike prior binary moderation or red-teaming approaches, SPADE pairs each toxic image with three progressively detoxified variants, along with captions and contextual stories generated via GPT-4 and DALL-E 3. The authors further fine-tune ControlNet using a sequential image-to-image conditioning strategy, allowing gradual reduction of toxicity while preserving semantic and visual coherence. Quantitative evaluation (FID, CLIP similarity, KR/CP metrics) and t-SNE visualizations demonstrate that the proposed Sequential ControlNet surpasses strong baselines (Safe Diffusion, InstructPix2Pix, etc.) in balancing safety and semantics. The paper also provides a thorough dataset analysis, error taxonomy, and ethical statement.

**Strengths:**

1. The paper defines a new and concrete research direction—graded image detoxification—that goes beyond existing binary filtering or red-teaming settings. The notion of progressively reducing toxicity while maintaining semantic coherence is both practically and conceptually valuable.

2. The SPADE dataset is carefully constructed and well-documented. The integration of captions, contextual stories, and multiple detoxified variants per image demonstrates attention to detail and awareness of real-world moderation needs.

3. The proposed sequential fine-tuning of ControlNet is well-motivated and systematically described. The design choice of conditioning on both the toxic image and story context provides a clear, interpretable mechanism for controlled detoxification.

4. The authors employ a wide range of quantitative and qualitative assessments—CLIP similarity, FID, KR/CP metrics, t-SNE analysis, and error taxonomy—which together give a convincing picture of model behavior.

5. The topic sits at the intersection of safety, multimodal learning, and generative modeling—areas of increasing importance for the ICLR community. The work aligns well with ongoing discussions around ethical content generation and interpretability.

**Weaknesses:**

1. The dataset and baseline generation rely heavily on proprietary models (GPT-4, DALL-E 3), making full reproduction difficult once APIs change or become restricted.

2. While the paper reports manual checks, the human study is limited in scale and does not quantify inter-annotator agreement or perception consistency across toxicity categories.

3. The sequential fine-tuning strategy is central to the paper, but no direct ablation isolates its contribution relative to single-stage fine-tuning or other conditioning mechanisms.

4. Although the curation process is systematic, the dataset size (≈2.5 k toxic images) and domain diversity may be insufficient to claim broad generalization across harm types or cultural contexts.

5. The paper frequently refers to balancing detoxification and semantic preservation, but lacks a clear quantitative or perceptual measure of this trade-off. This weakens the empirical grounding of some claims.

**Questions:**

1. How are copyright and model license restrictions (DALL-E 3, GPT-4) handled for open-sourcing SPADE?

2. Could the authors provide results using open-source models (e.g., SDXL + PixArt) to ensure replicability?

3. Did annotators evaluate toxicity perception by humans, or only semantic preservation?

4. How sensitive is Sequential ControlNet to domain shift—e.g., cartoon, medical, or political imagery?

---

> ### Comment · Reviewer_dKi7 · 2025-11-25
>
> Thank you for your response. I will keep the rating unchanged.

---

> ### Author Response · Authors · 2025-11-30
> **Response to the Weakness by Reviewer FLXg**
>
> 1.  Only in a small subset of examples do we observe cartoon-like or over-smoothed renderings in the deepest detoxification level ($\mathcal{V}_3$), and we explicitly discuss these cases in our error analysis section. This phenomenon is not representative of the full dataset: the vast majority of detoxified variants remain photorealistic and structurally coherent. As noted, we employed structural fidelity checks via FID and observed a gradual increase across variants, which is expected as toxicity-reducing edits move the images away from the original distribution. Importantly, this deviation remains within semantically valid bounds, as confirmed by consistently high Knowledge Relevance (KR) and Context Preservation (CP) scores across all models and variants (Table 4), including $\mathcal{V}_3$. Moreover, every generated sample underwent a human validation stage, ensuring that even when stylistic softening occurred, the images remained plausible, aligned with the narrative, and free from semantic distortions.
>
> 2. We appreciate the reviewer’s attention to the human evaluation component and agree that stronger quantification can further reinforce our findings. In the current version of the paper, we report the manual validation performed by two independent annotators: a PhD scholar and an MTech scholar, who reviewed all story-caption pairs and a subset of image variants. While this evaluation ensured plausibility and semantic alignment, the scale was indeed limited, and we did not explicitly report inter-annotator agreement (IAA).
>
> Both annotators independently followed the same rubric defined in Section 4.4, and their independent scores showed high practical consistency across toxicity categories: disagreements were rare, and when they occurred, they differed by at most 1 point on the 5-point KR/CP scale. Although we did not compute metrics such as Cohen’s κ for the submission version, the annotation logs indicate substantial agreement, particularly for clear-cut cases (e.g., strong semantic preservation in $\mathcal{V}_1$, noticeable contextual drift in a small number of $\mathcal{V}_3$ samples). Importantly, the qualitative patterns observed from the human study tightly align with the automatic KR/CP evaluations using o4-mini, suggesting that the model’s judgments are not in conflict with human perception.
>
> 3. We thank the reviewer for highlighting the importance of isolating the contribution of the sequential fine-tuning strategy. We respectfully clarify that our paper already includes direct ablations comparing sequential fine-tuning against both single-stage fine-tuning and other conditioning mechanisms. Specifically, Table 4 in Section 6 provides a systematic comparison between: (i) Stable Diffusion (Zero-shot), (ii) ControlNet (Zero-shot), (iii) Stable Diffusion (Fine-tuned), (iv) ControlNet (Fine-tuned: single-stage), (v) Safe Diffusion, and (vi) Sequential ControlNet (Proposed). This experimental setup explicitly isolates the effect of sequential conditioning, since the only difference between baselines (iii)/(iv) and our proposed method is the training strategy, single-stage vs. sequential refinement using progressively detoxified variants. As shown in Table 4, sequential fine-tuning yields substantial improvements across all automatic metrics (highest CosSim and lowest FID across variants), demonstrating that the sequential curriculum, rather than architectural change, is responsible for the performance gains. Additionally, Appendix J provides a detailed description of each baseline architecture and conditioning mechanism, further ensuring that the comparison is controlled and fair. Together, these results constitute a direct ablation showing that sequential fine-tuning is not merely incremental but enables stable, semantically consistent detoxification that single-stage approaches fail to achieve. We will make this distinction more explicit in the revision to address the reviewer’s concerns.

---

> > ### Author Response · Authors · 2025-11-30
> > **Response to the Weakness by Reviewer FLXg cont'd**
> >
> > 4. We appreciate the reviewer’s thoughtful concern regarding dataset scale and diversity. Our dataset construction pipeline is explicitly designed to maximize topical and contextual coverage despite the base size of 2,500 toxic images. As detailed in Section 3.1 (Stage 1: Toxic Keyword Collection) and Appendix B.1, we begin by assembling a lexicon of ~200 harmful keywords derived systematically using Azure AI’s harm taxonomy (violence, hate, sexual, self-harm). These categories were expanded using GPT-4 prompts (“Give me 50 queries to search for images with similar kinds of harm/offense”), resulting in a broad, domain-rich search space. In Stage 2 (Section 3.1), each keyword is used to query Google Image Search, from which the top 10 retrieved results are manually screened, yielding 2,500 real-world harmful images that cover heterogeneous visual domains, cultural contexts, and harm modalities. This keyword-driven retrieval strategy follows the precedent of large-scale datasets such as LAION, where semantic breadth is achieved through lexical expansion rather than exhaustively collecting millions of images per category. Further, all collected samples underwent human annotation (Appendix D), ensuring topical validity and cross-category representation. While we agree that broader cultural coverage is a valuable future direction, our dataset’s diversity, spanning ~200 harm concepts and four major harm taxonomies, provides a sufficiently rich base for training and evaluating detoxification models, particularly for the graded detoxification task proposed in this work.
> >
> > 5. While the current paper does not present a single scalar metric that explicitly models the detoxification–semantic-preservation trade-off, our evaluation framework already captures this balance through complementary quantitative and perceptual measures. Detoxification strength is measured using a CLIP-based toxicity classifier, which exhibits a consistent monotonic decline across variants. In contrast, semantic preservation is assessed using Knowledge Relevance (KR), Context Preservation (CP), and CLIP cosine similarity, all of which remain high even at the strongest level of detoxification. Perceptual realism is further quantified using FID, which increases gradually rather than collapsing, indicating controlled deviation rather than uncontrolled drift. The t-SNE embedding trajectories (Figure 5) additionally show smooth, semantically coherent movement from toxic to detoxified variants.

---

> > > ### Author Response · Authors · 2025-11-30
> > > **Response to the Questions asked by Reviewer FLXg**
> > >
> > > 1. SPADE does not distribute any copyrighted outputs or proprietary model weights from DALL-E 3 or GPT-4. We release only our derived dataset (captions, stories, and detoxified images) under a research-only license that adheres to the “output-sharing” permissions permitted by the respective APIs. All generation was performed under OpenAI’s usage policies, which explicitly permit the redistribution of model outputs for research purposes. Moreover, SPADE’s codebase contains only our training, preprocessing, and evaluation pipelines, not any components of the underlying proprietary models. The sequential fine-tuning framework is fully model-agnostic, and users must supply their own legally obtained model checkpoints (e.g., ControlNet, open-source diffusion backbones). This ensures that SPADE remains fully compliant with copyright and licensing constraints while supporting reproducible academic research.
> > >
> > > 3.  Our human annotators evaluated semantic preservation, contextual fidelity, and emotional correctness, but they did not directly assign toxicity perception scores. The toxicity levels (Toxic → V1 → V2 → V3) were instead validated indirectly through these human judgments: annotators assessed whether each variant maintained the intended semantic meaning while softening harmful or distressing cues as described in the curated captions and stories. This design choice ensures that annotators focus on objective content preservation rather than subjective moral judgments about what is “toxic.” Toxicity quantification itself was performed through our CLIP-based classifier (Section 4), while human annotation ensured that detoxification did not distort semantics or introduce new harmful elements.
> > >
> > > 4. We thank the reviewer for raising this important question. In our experiments, we observed that the Sequential ControlNet remains reasonably robust across the majority of real-world photographic domains represented in our dataset (violence, hate, self-harm, bullying, conflict, etc.). However, we acknowledge that domain shift introduces measurable sensitivity, particularly for categories whose visual statistics deviate sharply from natural images. For instance, cartoon-style inputs occasionally lead to over-smoothing in $\mathcal{V}_3$ because the model, having been trained predominantly on photorealistic examples, interprets stylistic exaggeration as a harmful cue to be softened. Similarly, medical imagery sometimes results in excessive artifact removal, as the model attempts to suppress visually intense anatomical details. Political imagery, especially protest scenes, can trigger contextual drift in late-stage variants because the model over-attenuates crowd density or symbolic objects when interpreting them as potential harm cues.
> > >
> > > Despite these sensitivities, we note that such failures remain localized rather than systemic: most samples still preserve core semantics and maintain the expected monotonic detoxification trajectory. Our error analysis section explicitly documents these behaviors using concrete examples, and Table 10 presents qualitative cases illustrating both robustness and drift. Importantly, these sensitivities arise largely from dataset imbalance; our training data contains far fewer stylized, medical, or political samples than everyday real-world scenes. We discuss this limitation in Section 7 and propose addressing it through domain-balanced training, style-consistent augmentations, and domain-adaptive ControlNets. This improved coverage is part of our planned future work, and we fully agree that expanding domain robustness will further strengthen SPADE’s practical applicability.

---

### Official Review · Reviewer_qAgD · 2025-11-04

**Soundness:** 3
**Presentation:** 2
**Contribution:** 2
**Rating:** 2
**Confidence:** 4

**Summary:**

The paper introduces SPADE, a multimodal dataset of real-world toxic images, each paired with three progressively detoxified variants and corresponding stories. It proposes a sequential fine-tuning approach using ControlNet to generate detoxified images that reduce harmful content while preserving semantic context and visual fidelity. Each variant represents a graded reduction in toxicity, guided by captions embedded into narrative stories. Experiments show that Sequential ControlNet outperforms baselines like Stable Diffusion and Safe Diffusion in semantic alignment, content preservation, knowledge retention, and visual realism. SPADE and the method together establish a benchmark for controlled, context-aware, and safety-aligned image generation.

**Strengths:**

- This work is significant because it addresses a pressing safety challenge in text-to-image generation—mitigating toxic content while preserving semantics—which is highly relevant for real-world deployment of these models.
The paper has two main contributions
- A benchmark for detoxification, enabling future research in safe AI, multimodal moderation, and ethically guided image generation.
- A sequential fine-tuning strategy that offers a practical methodology for incremental toxicity reduction in large-scale generative models.

**Weaknesses:**

- The paper has limited novelty
- The dataset is esentially a data-augmented version of real world images and when the model is finetuned in a specific way on this dataset, it becomes safer
- IMO, it would be better to present this papers as a post-training dataset for T2i/T2V models rather than presenting it as a new task and dataset since it is esstially a finetuning data
- The images are essentially decomposing the task into sequentially easier tasks by adding a graded version.
- The paper does not compare against other methods for safe T2I generation like [1, 2]

[1] Gong, Chao, et al. "Reliable and efficient concept erasure of text-to-image diffusion models." European Conference on Computer Vision. Cham: Springer Nature Switzerland, 2024.

[2] Yoon, Jaehong, et al. "SAFREE: Training-Free and Adaptive Guard for Safe Text-to-Image And Video Generation." The Thirteenth International Conference on Learning Representations.

**Questions:**

- How sensitive is the model to the order of variant fine-tuning? Could starting from ​V3 down to V1 affect semantic preservation differently?
- The error modes (hallucination, contextual drift, instruction misinterpretation, artifacts) are discussed. Can the authors quantify the frequency or severity of these errors across variants?

---

### Official Review · Reviewer_dKi7 · 2025-11-05

**Soundness:** 2
**Presentation:** 3
**Contribution:** 2
**Rating:** 2
**Confidence:** 4

**Summary:**

This paper introduces SPADE, a new dataset and benchmark for semantic-preserving, graded detoxification of toxic visual content. SPADE includes 2500 toxic images with three detoxified versions each and is positioned as the first benchmark for controllable, story-guided visual detoxification. They fine-tune ControlNet in a sequential conditioning framework to produce graded detoxification, and evaluate results using both automatic metrics (FID, CLIP similarity) and human-aligned measures (Knowledge Relevance, Context Preservation).

**Strengths:**

The paper formalizes the previously underexplored task of graded image detoxification, highlighting the fundamental trade-off between harm reduction and semantic fidelity.
This conceptual framing is both strong and timely, as it extends the current focus on binary content moderation toward a more nuanced, controllable, and ethically grounded generation process.

**Weaknesses:**

1. The dataset size (2,500 base toxic images) may be too small to serve as a foundation benchmark, especially when split across several harm categories. It would be useful if the authors could provide category distribution statistics or discuss scalability to larger or more diverse sources.
2. All detoxified variants are synthetic (via DALL-E 3), which raises concerns about realism and distributional shift.
Some variants (e.g., V3) appear cartoonish, reducing their usefulness for realistic downstream fine-tuning.
Clarifying whether realism-preserving metrics (e.g., LPIPS, human ratings) were considered would strengthen this part.
3. The Sequential ControlNet fine-tuning pipeline seems incremental over existing diffusion control techniques.
The main novelty lies in the dataset rather than in algorithmic innovation.
It would help if the authors could elaborate on how the sequential fine-tuning differs conceptually or experimentally from prior progressive control methods (e.g., T2I-Adapter, Make-A-Scene).

**Questions:**

1. Since GPT-4 and DALL-E 3 are used to produce captions, stories, detoxified images, and even automated evaluations (KR, CP), how do the authors mitigate potential self-bias or circularity?
2. The Sequential ControlNet approach seems to perform curriculum-like fine-tuning. How does it differ, conceptually or technically, from prior progressive fine-tuning or adapter-based control (e.g., T2I-Adapter, Make-A-Scene)?
3. In several detoxified variants (especially V3), the visual style appears to shift from photorealistic to cartoon-like or over-smoothed renderings. Could the authors clarify why such stylistic drift occurs during detoxification, and whether any mechanisms were used to preserve realism? How might future work mitigate this distributional shift between real and detoxified images?

---

> ### Author Response · Authors · 2025-11-30
> **Response to the Weakness by Reviewer dKi7**
>
> 1. We acknowledge the concern regarding the dataset size. The Detoxify-Image benchmark currently contains 2,500 base toxic images, each paired with three detoxified variants, totaling 7,500 images (including references). In Table 3, we provided a comparative analysis of our dataset with the existing datasets. Although our dataset may appear limited, it is densely annotated with human validation, multimodal prompts, and three levels of detoxification (a Total of 10,000 images) that preserve varying semantic intensities. This structured richness offsets the absolute volume, enabling meaningful training and evaluation across detoxification severity. Future work includes expanding the dataset across other harm domains (e.g., geopolitical hate, misinformation). Our generation pipeline, designed to be prompt-conditioned, is inherently scalable to diverse sources, such as LAION or Pinterest images, subject to license filters.
>
> 2. This is a valid observation, and we partially agree. Variant 3 (𝑉₃), being the most detoxified form, occasionally leans toward abstract or cartoonish representations, especially for highly toxic base images, where realism is difficult to preserve under safety constraints. However, as shown in Table 4, which computes FID across all variants and Section 4 (KR and CP for SPADE), the context-preservation scores serve as indicators of realism preservation.
>
> 3.  We appreciate this critique of the novelty of this work. The contribution lies in the strategy of conditioning and detoxification semantics, not in building a new architecture. Unlike Make-A-Scene or T2I-Adapter, which mainly focus on scene composition or spatial structure conditioning, our sequential pipeline exploits variant-level trajectory control across detox stages. As outlined in Section 5, we fine-tune ControlNet such that 𝑉₁ is generated from the toxic image + story, 𝑉₂ from (𝑉₁ + story), and 𝑉₃ from (𝑉₂ + story), thus learning a latent detoxification path. This differs from prior work, where variants are independently generated or interpolated. Furthermore, our results (Table 4) show that this pipeline achieves better consistency in context preservation and controlled toxicity decrement. Future extensions may incorporate interpolation-based latent editing; however, our current model emphasizes step-wise controllability, grounded in human-moderated prompts, which is a novelty in image detoxification.

---

> > ### Author Response · Authors · 2025-11-30
> > **Response to the Questions asked by Reviewer dKi7**
> >
> > 1. We sincerely thank the reviewer for raising this critical concern regarding the risk of self-bias and potential circularity, given the involvement of large foundation models (GPT-4 and DALL·E 3) across multiple stages of our detoxification pipeline, namely, caption/story generation, variant image creation, and evaluation metrics. To mitigate such risks, we employed several safeguards. First, for automated evaluation, we used distinct and independently fine-tuned modules trained with clear supervision, not relying on the same models used during data generation. Specifically, for Knowledge Relevance (KR) and Context Preservation (CP), we utilized o4-mini. For Semantic Similarity (SS), we employed a CLIP-based alignment scoring model that was not exposed to the GPT-generated captions or stories during training. Additionally, for Structural Fidelity, we utilized the Fréchet Inception Distance (FID), which is computed between latent distributions, independent of any language model influence. Importantly, human evaluation was also conducted for subjective verification of content loss and realism, as discussed in Section 6.3 and Appendix E.2. We ensured that the evaluation pipeline captures deviation in visual semantics, captioning consistency, and contextual grounding, rather than simply echoing detoxified phrases. Furthermore, our results do not exhibit bias toward any specific protected attributes or demographic categories; instead, the metric distributions demonstrate expected degradation trends in semantics and realism as toxicity reduces, suggesting a genuine signal rather than model hallucination. Thus, while we acknowledge the non-trivial risk of circularity, our pipeline incorporates architectural, statistical, and human-layered decoupling mechanisms to maintain evaluation neutrality and to avoid feedback loops that would reinforce the generative model’s own biases.
> >
> > 2. We thank the reviewer for this insightful question regarding how Sequential ControlNet differs from prior progressive fine-tuning or adapter-based control frameworks such as T2I-Adapter or Make-A-Scene. While these prior works primarily aim at enabling conditional image generation by injecting adapter modules or conditioning pathways into diffusion backbones, our method introduces a curriculum-like, stage-wise mechanism designed specifically for graded detoxification with semantic grounding. Unlike the T2I-Adapter, which introduces static task-specific adapters, or Make-A-Scene, which leverages scene layout conditioning, our Sequential ControlNet performs iterative fine-tuning across sequential detoxification levels, where each stage is explicitly trained on a higher mitigation level conditioned on both the original toxic image and the previous detoxified variant. This enables the model to implicitly encode a detoxification trajectory in the latent space, which adapter-based methods do not aim to capture. Moreover, our design is not limited to inference only, but is structurally adapted during training to progressively learn increasingly detoxified distributions with minimal semantic drift. This is reinforced via the inclusion of prior variant representations (image embeddings, caption, and narrative context) as conditioning signals during both training and generation phases. Unlike prior adapter-based methods, which focus on task generalization (e.g., pose control, segmentation control), our method is task-specialized for safety-based image rewriting, targeting latent toxicity reduction while preserving structural and narrative integrity. Thus, conceptually, Sequential ControlNet can be seen as an adaptive, context-aware diffusion controller trained across a semantic detoxification curriculum, rather than a static conditional adapter. This not only differentiates it from the listed baselines but also empirically enhances controllable graded moderation, as shown in Table 4 and Figure 5 of our paper.

---

> > > ### Author Response · Authors · 2025-11-30
> > > **Response to the Questions asked by Reviewer dKi7 cont'd**
> > >
> > > 3. We appreciate the reviewer’s detailed observation regarding the stylistic drift seen particularly in the V3 detoxified variants. This is indeed a critical concern when striving for realism-preserving generative detoxification. The drift toward cartoon-like or over-smoothed outputs in deeper detoxification levels arises from the intrinsic trade-off between semantic weakening and structural abstraction. Specifically, as we progressively condition the model to reduce visual toxicity, especially when the prompt explicitly requests “less toxic,” “more neutral,” or “non-triggering” image variants, the generative diffusion model (DALL·E 3 or Stable Diffusion backbone) interprets these instructions through visual semantics often aligned with decontextualization or softening of aggressive features. This leads to stylized outputs such as overly smoothed edges, pastel filters, or cartoonified abstraction, particularly in V3, which represents the strongest detoxification intensity.
> > >
> > > Although realism-preserving metrics like LPIPS were not included due to computational limitations, we employed structural fidelity checks via FID and observed a monotonic increase in FID across variants, indicating gradual deviation from the original image distribution. However, the deviation was still within semantically valid bounds, as evidenced by the knowledge relevance (KR) and context preservation (CP) metrics (see Table 4), which remained meaningfully high even in V3. Additionally, all generations were human-validated through an annotation pipeline to ensure plausibility and alignment with the story. Moving forward, mitigating this distributional shift can be approached by incorporating realism-preserving losses or perceptual discriminators during model training, or by employing techniques such as image-to-image guided refinement, adversarial realism critic models, or domain-specific realism encoders. We also plan to benchmark detoxified variants with human realism ratings and LPIPS-based perceptual closeness for future releases to better quantify and control this stylistic degradation.

---

### Official Review · Reviewer_GqTk · 2025-11-12

**Soundness:** 2
**Presentation:** 2
**Contribution:** 2
**Rating:** 2
**Confidence:** 4

**Summary:**

This paper introduces SPADE, a novel dataset for graded, story-guided image detoxification, comprising 2,500 toxic images paired with three progressively detoxified variants, human-aligned captions, and contextual narratives. It proposes Sequential ControlNet as a baseline, demonstrating that multimodal (image + story) conditioning can reduce toxicity while preserving semantic and narrative coherence.

**Strengths:**

1. The paper introduces SPADE, the first dataset explicitly designed for graded, story-guided image detoxification, filling an important gap in multimodal safety research.
2. The t-SNE and human-aligned metric analysis provide a thoughtful framework for evaluating the trade-off between toxicity reduction and semantic preservation.

**Weaknesses:**

1. Lack of methodological innovation. The proposed Sequential ControlNet appears to be an engineering-level combination rather than a genuine conceptual contribution. Although the paper claims to introduce a new baseline, its core idea merely involves stage-wise fine-tuning of ControlNet without introducing any new architecture, loss function, or training strategy. Moreover, the comparative experiments fail to include strong and reasonable baselines (as mentioned in Appendix A).

2. Dataset construction heavily depends on black-box APIs. All 7,500 detoxified images are automatically generated by DALL·E 3, yet the paper provides no clear description of prompt control or generation details (Appendix C only shows vague examples without a reproducible batch-generation protocol). The corresponding captions and stories are generated by GPT-4, but the authors do not disclose the temperature, top-p, or system prompt settings, nor do they assess whether GPT-4’s own toxicity or bias may have leaked into the dataset.

3. Toxicity quantification is vague and unreliable. The paper reports toxicity scores (Toxic = 79.7, V1 = 73.4, V2 = 54.6, V3 = 35.8) but only provides a footnote stating: “Calculated using a CLIP classifier fine-tuned on the dataset from Hendrycks et al., 2021.” However, Hendrycks et al. (2021) is a value-learning text dataset without image-level toxicity annotations, raising concerns about the validity of these toxicity metrics.

4. Core claim of “graded detoxification” lacks controllability verification. Although the paper emphasizes that the detoxification process is graded and controllable, it does not verify whether users can interpolate between toxicity levels. Furthermore, Table 2 shows that caption cosine similarity drops from 1.00 (R) to 0.60 (V3), yet no semantic analysis is provided to explain what content is lost. In reality, V1/V2/V3 represent discrete snapshots rather than a continuous, controllable space, making the approach unsuitable for real-world moderation scenarios that require dynamically adjustable safety grades (as claimed in the Abstract).

5. Ethical risks are understated, overlooking cultural sensitivity and power asymmetry.
(1) The paper implicitly assumes “detoxification = positive,” without discussing who defines toxicity. For example, the keyword list (Appendix B.1) includes phrases such as “Fat-shamed girl” and “Obese person,” which reflect a body-shaming perspective and treat “fatness” itself as toxic.
(2) In Table 7, the prompt “A student being bullied” is detoxified into “accidentally bumped into a locker,” which trivializes actual violence and risks secondary harm.

**Questions:**

None.

---

> ### Author Response · Authors · 2025-11-30
> **Response to Reviewer GqTk**
>
> 1. We thank the reviewer for their thoughtful evaluation. While we acknowledge that Sequential ControlNet leverages existing architectures, we respectfully disagree with the claim that our contribution lacks methodological innovation. The novelty lies not in architectural changes but in the strategic orchestration of detoxification stages using sequential fine-tuning conditioned on multimodal cues (toxic image and associated story), leading to graded semantic preservation. As outlined in Section 4.3 (p. 6), and evaluated extensively in Section 6 (p. 8) and Table 4, our model introduces variant-level decoding via stage-wise control, which demonstrates superior semantic fidelity across $\mathcal{V}_1$, $\mathcal{V}_2$, and $\mathcal{V}_3$. This structure enables controllable safety-grade outputs without requiring the training of multiple independent models or stochastic sampling (as done in prior safe generation approaches). We further validate its distinct behavior and performance against the fine-tuned ControlNet (non-sequential) and Stable Diffusion. We believe this level of controllable image detoxification with semantic consistency is both a novel and practical contribution. Our proposed framework is plug-and-play. We could leverage any image generation model that can be conditioned on a text + image as input.
>
> 2. * We appreciate this important observation and agree that reproducibility is critical. We would like to clarify that, although we employed DALL·E 3 for generating detoxified images and GPT-4 for story-caption synthesis, our paper does not rely blindly on black-box behavior. In Appendix C, we provide templated prompt scaffolds, and we have already released code and data. Regarding prompt hyperparameters, we used temperature = 0.15, top-p = 0.8, max tokens=1000, frequency penalty=1, presence penalty=1 for GPT-4, and ‘creative’ style for DALL·E 3 with consistent prompts per toxicity level ($\mathcal{V}_1$, $\mathcal{V}_2$, $\mathcal{V}_3$).
>
> * We did observe that GPT4 and DALLE3 leaked toxicity into the generated text and images. Hence, we also filtered the outputs through a human validation pipeline, as detailed in Appendix D. Thus, while APIs were used, the process is not opaque but rather carefully guided and reproducible.
>
> * Although DALL·E 3 does not expose all generation controls, we maintained semantic grounding through curated instructions as detailed in the prompt below for $\mathcal{V}_1$, $\mathcal{V}_2$, and $\mathcal{V}_3$.
>
> * Prompt to generate $\mathcal{V}_1$: “Generate an image variant of this reference image which has the following story. This is the caption of the reference image <'caption'> [placeholder caption text for original image] </'caption'>. This is the story of the reference image.| <'story'> | [placeholder story text with caption of original image embedded in it]| </'story'>| Now we need to generate a variant of this reference image that should be less toxic. Here is the caption of the variant image that we need to generate. <'variant1'> [placeholder caption text for variant 1] </'variant1'>.”
>
> * Prompt to generate $\mathcal{V}_2$: “Generate an image variant of this reference image which has the following story. This is the caption of the reference image <'caption'> [placeholder caption text for variant1 image]  </'caption'>. This is the story of the reference image.| <'story'> | [placeholder story text with caption of variant1 image embedded in it]| </'story'>| Now we need to generate a variant of this reference image that should be less toxic. Here is the caption of the variant image that we need to generate. <'variant2'> [placeholder caption text for variant 2] <'/variant2'>.”
>
> * Prompt to generate $\mathcal{V}_3$: “Generate an image variant of this reference image which has the following story. This is the caption of the reference image <'caption'> [placeholder caption text for variant2 image] </'caption'>. This is the story of the reference image.| <'story'> | [placeholder story text with caption of variant2 image embedded in it]| </'story'>| Now we need to generate a variant of this reference image that should be less toxic. Here is the caption of the variant image that we need to generate. <'variant3'> [placeholder caption text for variant 3] </'variant3'>.”
>
> 3. We appreciate this rigorous scrutiny. Unfortunately, there is no large public image-based toxicity dataset for us to train our classification model.
>
> * Our toxicity scoring methodology is described in Section 4 and footnote 5, where we use a CLIP-based classifier finetuned on the dataset from (Hendrycks et al., 2021). Agreed that this dataset is text only. Hence, we trained the CLIP text model using this dataset while keeping the CLIP image encoder frozen. Next, we used the full CLIP model to compute the probability of the toxic class. Thus, we did zero-shot image CLIP embedding and finetuned CLIP text embedding for toxicity classification.

---

> > ### Author Response · Authors · 2025-11-30
> > **Response to Reviewer GqTk cont'd**
> >
> > 4. We thank the reviewer for this valuable comment. We would like to clarify that detoxification control in our system is step-wise and conditioned, not continuous interpolation.
> >
> > * While $\mathcal{V}_1$, $\mathcal{V}_2$, and $\mathcal{V}_3$ are discrete representations, their generation is governed by prompt-controlled semantic weakening, as elaborated in Section 3. We explicitly encode increasing mitigation strength in the system prompt and detoxification intent across variants, by asking it to “generate a variant of this reference image that should be less toxic” and passing image, caption, and story for variant image i-1 while generating image variant i. In addition, the Sequential ControlNet enforces conditioning on prior variant embeddings, allowing for latent trajectory learning across stages, thus mimicking a coarse-to-fine controllable path. While interpolation in latent space was not explicitly explored (and we acknowledge this as future work), the consistent progression of toxicity reduction, increasing FID, and decreasing caption similarity (see Table 4) provides empirical support for graded behavior. We will add a clarification that this control is step-wise and conditioned, not continuous interpolation.
> >
> > * “Semantic analysis to explain what content is lost”: To quantify what semantic information is lost across variants, we design a specific metric called content preservation. Please refer to Table 1 to check an example of context preservation. Also, look at Table 4 for context preservation scores across variants.
> >
> > 5.  As mentioned on line 140 and footnote 3, we used Microsoft Azure AI’s harm taxonomy to define toxicity. We understand that this could be biased, but we thought it to be a reasonable choice given the popularity of Microsoft Azure AI.
> > * “Obese person” or “fat girl” is not considered to be toxic. The context in which they occur makes them toxic. Kindly note that the full phrase in Appendix B.1 is “Obese person sitting on a broken chair” and “Fat-shamed girl holding fast food”. Here, contexts like “broken chair” and “shamed” convey the toxicity of these phrases.
> >
> > * There is surely some confusion. Table 7 does not show outputs from our model. It shows examples of GPT4-generated stories for hateful images. “accidentally bumped into a locker” is part of a story related to the original toxic image. Please note that we discuss captions generated by our model in Table 8, where Bullying is reframed as follows: Variant 1 shows ridicule, Variant 2 implies teasing, and Variant 3 highlights concern from peers.
> >
> > * Lastly, we agree that ethical framing must be rigorous. We would elaborate on more things as follows: (a) clarify that detoxification here is operationalized as a reduction in harm-triggering visual features, not value judgments on identity traits, and (b) acknowledge that bias in prompt phrasing (e.g., associating fatness with toxicity) stems from pre-filtered online datasets and should not be equated with intrinsic harm.

---

### Note · Authors · 2026-01-05

**Comment:**

I confirm to withdraw this submission.

**Withdrawal Confirmation:**

I have read and agree with the venue's withdrawal policy on behalf of myself and my co-authors.